# Time Series Domain Adaptation via Channel-Selective Representation Alignment

**Nauman Ahad**                                                                    *nahad3@gatech.edu*
*School of Electrical & Computer Engineering*
*Georgia Institute of Technology*

**Mark A. Davenport**                                                              *mdav@gatech.edu*
*School of Electrical & Computer Engineering*
*Georgia Institute of Technology*

**Eva L. Dyer**                                                                    *evadyer@gatech.edu*
*Department of Biomedical Engineering*
*Georgia Institute of Technology*

**Reviewed on OpenReview:** *https://openreview.net/forum?id=8C8LJIqF4y*

## Abstract

Building generalizable and robust multivariate time series models can be challenging for real-world settings that involve significant shifts between training and testing. Existing unsupervised domain adaptation methods often struggle with real world distribution shifts which are often much more severe in some channels than others. To overcome these obstacles, we introduce a novel method called *Signal Selection and Screening via Sinkhorn alignment for Time Series domain Adaptation* (SSSS-TSA). SSSS-TSA addresses channel-level variations by aligning both individual channel representations and selectively weighted combined channel representations. This dual alignment strategy based on channel selection not only ensures effective adaptation to new domains but also maintains robustness in scenarios with training and testing set shifts or when certain channels are absent or corrupted. We evaluate our method on several time-series classification benchmarks and find that it consistently improves performance over existing methods. These results demonstrate the importance of adaptively selecting and screening different channels to enable more effective alignment across domains. Python implementation of our method can be found at https://github.com/nerdslab/SSSS_TSA.

## 1 Introduction

Time series analysis is increasingly pivotal in diverse fields such as astronomy, climate science, neuroscience, healthcare, finance, and industrial monitoring (Baker et al., 2015; Bock et al., 2021; Amjad & Shah, 2017; Coelho et al., 2022). However, due to a variety of factors including drift, sensor differences, and measurement limitations, there can often be significant shifts in the data between the training and testing times (He et al., 2023). Traditional methods often struggle with the variability inherent in time-series data, leading to suboptimal performance and limited generalization. This challenge underscores the need for more robust and adaptable models that can effectively manage these complexities and leverage the full potential of time-series data.

Recent work has shown the promise of domain adaptation approaches for time series (Ragab et al., 2023; He et al., 2023; Liu & Xue, 2021) to help address some of these challenges. In this setting, we combine labels on the training set with unlabeled test data to build a unified encoder even in light of significant shift across the two sets. While these methods perform well when tested in some types of domain shift, in the context of

multivariate (multi-channel) time series, these models fail when presented with missing channels or when there are significant shifts in individual channels.

Channel-level shifts are surprisingly commonplace and yet haven't been accounted for in the existing domain adaptation literature. Take for instance an example from a human activity recognition datasets in Figure 1, where class 1 (sitting – blue) and class 2 (standing – red) need to be adapted from the source domain shown in purple to target domain in yellow. In Fig. 1(A), a large domain shift in the blue channel can cause the representations of the source and target classes to be misaligned which can result in disastrous performance on the target domain. If this blue channel is simply ignored as in Fig. 1B, the target representations are much likely to align with their respective classes in the source domain, resulting in greatly improved performance in the target domain. This example emphasizes the need to develop domain adaptation methods that account for shifts in each channel differently while also having the ability to screen channels involving larger domain shifts.

To address these challenges, we introduce a novel approach for time series domain adaptation. Our approach centers on constructing a *separable alignment plan* between the labeled (source) and unlabeled (target) data, where the goal is to first *align each channel* and then *align the joint embeddings* formed after pooling across channels. To achieve a sparse and selective attention of channels when pooling, we employ a simpler variant of self-attention to select and combine channels, enabling the fusion of the channel latent representations into a comprehensive global representation. This method not only enhances adaptability across domains but also allows for discernment in channel selection and screening, ensuring that only the most relevant and informative channels are utilized for alignment and inference.

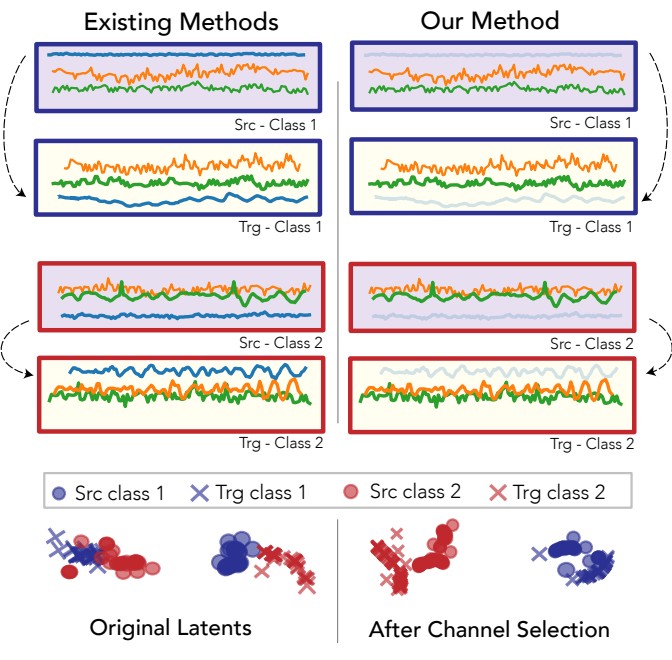

Figure 1: **Examples of channel-level shifts.**

We evaluate the performance of the model on a number of time series benchmarks and achieve state-of-the-art performance. In benchmark tests on a widely used human activity recognition dataset called WISDM, our method achieved a nearly 6% improvement over the existing state-of-the-art models. Our results and ablations not only demonstrate the effectiveness of our approach in dealing with complex, multi-channel time-series data but also highlights its potential in identifying the most informative channels across two datasets for diagnostic and interpretation purposes.

Our contributions include:

- **Novel method for time series domain adaptation.** We develop a new method that builds a separable alignment plan that *aligns each channel* independently before pooling across channels and *aligning the fused representations.*

- **Channel selection and screening via a self-attention layer.** Our method employs a self-attention layer for sparse and selective attention of channels. This allows for the efficient selection and combination of only the most relevant channels, leading to the formation of a robust global representation that is representative of the essential features in the data even in the context of significant shifts in some channels (see Fig. 1).

- **State-of-the-art performance in time-series classification benchmarks.** Our approach achieves state-of-the-art performance on a number of datasets, and achieves a 6% improvement in accuracy over existing state-of-the-art methods on the WISDM human activity dataset. This result

underscores the effectiveness of our method in handling shifts in complex, multi-channel time-series data.

- **Interpretability.** Our approach not only excels in performance but also provides insights into the most informative channels across datasets. This feature is particularly beneficial for diagnostic purposes, allowing practitioners to understand which channels provide the most joint information across the source and target.

## 2 Background and Related Work

### 2.1 Domain adaptation

Many real-world scenarios require adapting a model which is trained on a source labeled dataset to a *related* unlabeled target dataset. This related dataset can have a shift in either the unlabeled data (feature shift), or the (unavailable) labels in the target domain.

Domain adaptation methods try to improve prediction performance on unlabelled target domain data by leveraging source domain labeled data. Most methods addressing feature shift, which is the domain shift we address in our paper, make the assumption that the shifted class conditioned data in the target domain is closer to the corresponding class conditioned source data in the representation space. This means that the source domain representations for a class should be closer to the target domain representations of the same class than the target representations of other classes (Ben-David et al., 2010; van Tulder & de Bruijne, 2023; Zhao et al., 2019). When this assumption is met, making the source and target domain representations *invariant* while simultaneously minimizing the source classification loss can help models adapt to target domains.

These source and target representations can be made invariant through adversarial learning (Ganin et al., 2016; Long et al., 2018), or minimizing distances such as the maximum mean discrepancy or Wasserstein distance (Wang et al., 2023a; Shen et al., 2018; Sun & Saenko, 2016; Damodaran et al., 2018). Other methods take an alternate approach where source domain labels are used to generate pseudo labels in the target domain, which are then used to train a model to classify unlabeled target domain data (Wang & Breckon, 2020). Related methods have incorporated augmentations with contrastive learning on both source and domain representations to better adapt models to the target domain (Singh, 2021).

### 2.2 Domain adaptation techniques in time series

Time series domain adaptation methods have mostly adopted methods from vision while utilizing encoders more suited to time-series data such as RNNs and 1D temporal CNNs. Many methods utilize adversarial learning (Wilson et al., 2020), or use kernels more suited to time-series data to align source and target representations (Liu & Xue, 2021; Cai et al., 2021). Other methods such as SASA make the assumption that inter channel relationship is invariant across domains (Cai et al., 2021). Recent benchmarking results suggest that this assumption isn't true in time series domain adaptation settings where larger shifts often occur in a few channels across domains(Ragab et al., 2023). Recent methods have also additionally incorporated pseudo-labeling and self-supervised contrastive learning through augmentations (Ragab et al., 2022; Ozyurt et al., 2023). As frequency domain information can be in some cases useful for time series classification tasks, newer methods have also incorporated source and target domain frequency representations while learning domain invariant representations (He et al., 2023).

All of these methods pass all time series channels collectively into a common neural network encoder. There has been very little, to the best of our knowledge, that develops different representations and domain adaptation schemes for each channel in multivariate time series. There is only work that we know of utilizes individual channel alignment (Wang et al., 2023b). The motivation for this method is to develop improved domain adaptation for sensor networks which require localized channel/sensor level adaptation . Our method on the other hand utilizes a signal selection layer to downweigh channels with significant shifts between the source and the target domain which requires first aligning individual channel representations as we will discuss in section 3.2.

### 2.3 Optimal Transport and Sinkhorn divergences

To build aligned representations, we use the Sinkhorn divergence, a robust measure of distributional similarity. The Sinkhorn divergence is an entropic regularized variant of Wasserstein distances. The entropic regularization, with regularization parameter $\gamma$, allows a computationally efficient transport plan solution to be obtained through Sinkhorn iterations (Cuturi, 2013). This divergence can then be used to measure how similar two different samples sets are, and thus can be used as a loss function to make two sample sets similar (Genevay et al., 2018). We provide more details for Sinkhorn divergence in Appendix A.

## 3 Method

### 3.1 Motivation

Many real-world time series classification problems often heavily depend on a limited number of channels, and the quality of different channels may vary between training and testing times. This variability can lead to a significant drop in model performance when the channels' characteristics change, highlighting the necessity for models that can dynamically adapt to such changes (He et al., 2023). Building robust models that can achieve channel-level alignment is thus critical for improving generalization in time series data.

Motivated by these challenges, we propose a novel approach for time-series domain adaptation that addresses the issue of selective channel-level alignment. Our method decouples the alignment processes across channels, focusing specifically on aligning channels at a granular level as well as the alignment of the mixture of channels. This approach ensures that the most informative and stable channels are prioritized, while less informative or unstable channels are effectively de-emphasized during alignment.

Figure 2 illustrates the overall architecture of our proposed method. Initially, the time series data is split into different channels, with each channel being fed into a specific encoder and classifier. The individual channel representations are then processed to obtain channel weighting scores. These scores are used to reweight the channel representations, effectively selecting certain channels to highlight and others to mask. This selection and screening process ensures that only the most relevant channels contribute significantly to the final model.

Our model leverages a channel-level encoder and a signal selection mechanism to reweight different channel representations before aligning the source and target domains. This reweighting is crucial for dynamically adjusting the importance of each channel based on its relevance and stability, thereby enhancing the model's robustness and adaptability. Overall, our approach addresses the core issues of channel variability and misalignment, providing a more resilient solution for time series classification in real-world scenarios. By focusing on channel-level alignment and reweighting, our method significantly improves the generalization and adaptability of the model across different domains.

### 3.2 Method description

Let $\boldsymbol{X}_t \in \mathbb{R}^{C \times T}$ denote the target data, a multivariate time-series dataset with length $T$, and $C$ channels. Similarly, we will consider the source timeseries $\boldsymbol{X}_s \in \mathbb{R}^{C \times T}$ which could be of different number of timepoints or channels than the target dataset.

**Building channel-level representations.** In line with recent work in timeseries classification and forecasting which show state-of-the-art performance when separating timeseries into univariate single-channel information before mixing across channels (Liu et al., 2022; Nie et al., 2022; Tao et al., 2020), we first split the dataset into $C$ different, univariate time series $\boldsymbol{x}^c \in \mathbb{R}^T$, where the superscript $c$ represents the $c^{\text{th}}$ channel. Each of these univariate time series is then fed into a channel specific encoder $f_\theta^c$, to obtain channel specific representations $\boldsymbol{z}^c = f_\theta^c(\boldsymbol{x}^c) \in \mathbb{R}^d$. These encoders are used to encode both source domain time series $\boldsymbol{X}_s$ and target domain time series $\boldsymbol{X}_t$ into $\boldsymbol{z}_s^c$ and $\boldsymbol{z}_t^c$ for all channels $c \in C$. The source domain representations for the channels, $\boldsymbol{z}_s^c$, are then linearly transformed by $\boldsymbol{W}_c \in \mathbb{R}^{d \times M}$ for all $c \in C$, where $M$ is the number of classes. A softmax function is then applied on this linear transformation to produce the class prediction vector $\hat{\boldsymbol{y}}_s^c$.

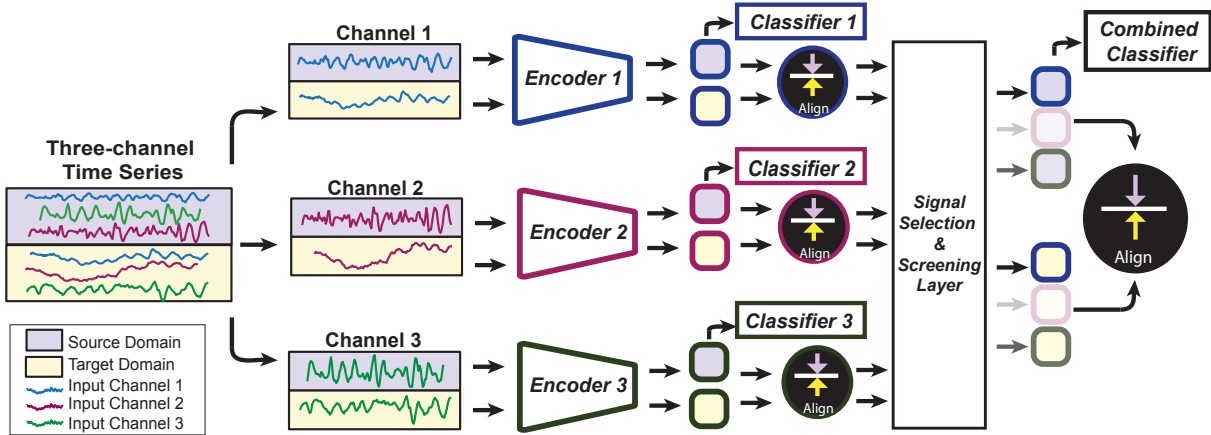

Figure 2: **Overview of our proposed approach.** For both the source and the target domain time series, each of the input channels is fed into channel specific encoders before minimizing source classification and alignment loss. All of the domain adapted channel representations are then provided to a channel selection layer, which reweights source and target channel representations before passing all of the channels through to minimize the subsequent alignment and classification losses.

**Channel screening and selection.** To help deal with large amounts of shift between source and target domains, we develop a screening procedure which computes a weighting vector $\boldsymbol{w}$ to be applied to the channel-level information. The channel-level weights can be obtained by:

$$\boldsymbol{w} = \text{softmax}\left(\frac{1}{\tau}\left[\frac{1}{\sqrt{d}}\left((\boldsymbol{q}^1)^\intercal \boldsymbol{k}^1, \dots, (\boldsymbol{q}^C)^\intercal \boldsymbol{k}^C\right)\right]\right) \tag{1}$$

where $\boldsymbol{k}^c = \boldsymbol{K}\boldsymbol{z}_c$ and $\boldsymbol{q}^c = \boldsymbol{Q}\boldsymbol{z}_c$ $\boldsymbol{K}$ and $\boldsymbol{Q} \in \mathbb{R}^{d \times d}$, and $\tau$ is the softmax scaling factor. This scaling is used to reweight our channel-independent representation as $\boldsymbol{z}^a \in \mathbb{R}^{dC}$:

$$\boldsymbol{z}^a = \text{vec}\left(\boldsymbol{w} \odot \boldsymbol{Z}\right) = \text{vec}\left([w^1 \boldsymbol{z}^1, w^2 \boldsymbol{z}^2, ..., w^C \boldsymbol{z}^C]\right).$$

Figure A1 in the appendix provides a visual description of this channel screening and selection method.

**Overall loss.** Our overall alignment and classification loss is a function of all channel encoders ($f_\theta^c$) and channel classifiers ($\boldsymbol{W}^c$):

$$\mathcal{L} = \underbrace{S_\gamma(\boldsymbol{z}_s^a, \boldsymbol{z}_t^a) + \mathcal{L}_{CE}(\hat{\boldsymbol{y}}_s^a, \boldsymbol{y})}_{\text{Dom. adapt. for combined reps}} + \sum_{c=1}^{C} \underbrace{S_\gamma(\boldsymbol{z}_s^c, \boldsymbol{z}_t^c) + \mathcal{L}_{CE}(\hat{\boldsymbol{y}}_s^c, \boldsymbol{y}_s)}_{\text{Dom. adapt. for each channel}}, \tag{2}$$

where we use the Sinkhorn distance $S_\gamma$ with entropy parameter $\gamma$ as defined in Eq. 3 to measure the deviation between two distributions due to its robustness and ease of use. See the Appendix A for a full discussion of the Sinkhorn algorithm.

### 3.3 Intuition behind our method

Different time series channels can carry more diverse information than channels within other modalities such as images. As a result, many real-world time series classification problems can often heavily depend on a limited number of channels. Learning separate classifiers for each channel, which is required as part of our method, leads to individual channel representations that try to maximize the mutual information between each channel input and source label data. When the supervised loss on the source domain is combined with a loss that minimizes the Sinkhorn distance between source and target data for each channel, the signal selection and screening layer now produces weights that not only down weigh channels that are uninformative for classification in the source domain, but also de-emphasize channels that do not align well between the source and target domains. As we will see, this can ultimately significantly improve domain adaptation performance.

Table 1: **Mean accuracy and macro F1 scores on timeseries domain adaptation benchmarks over 5 runs (↑ is better). Standard deviation of these scores is provided in Appendix table A15.**

| Method | Mean Shift | | UCIHAR | | HHAR | | PXECG | | WISDM | | WISDM-Bal | |
|---|---|---|---|---|---|---|---|---|---|---|---|---|
| | ACC | F1 | ACC | F1 | ACC | F1 | ACC | F1 | ACC | F1 | ACC | F1 |
| Sup | 43.12 | 0.423 | 77.04 | 0.750 | 59.40 | 0.543 | 63.51 | 0.366 | 64.90 | 0.504 | 65.84 | 0.521 |
| DANN | 71.32 | 0.701 | 82.91 | 0.857 | 71.27 | 0.678 | 62.87 | 0.347 | 67.94 | 0.567 | 73.86 | 0.683 |
| AdvSKM | 74.31 | 0.712 | 85.12 | 0.813 | 63.25 | 0.616 | 62.98 | 0.372 | 69.92 | 0.581 | 71.19 | 0.611 |
| CoDATS | 54.31 | 0.531 | 86.34 | 0.856 | 68.79 | 0.686 | 66.30 | 0.366 | 68.35 | 0.548 | 75.15 | 0.665 |
| CDAN | 79.54 | 0.813 | 84.59 | 0.836 | 70.06 | 0.704 | 64.29 | 0.375 | 70.12 | 0.517 | 70.29 | 0.661 |
| SASA | 63.72 | 0.587 | 80.75 | 0.791 | 65.85 | 0.641 | **66.47** | 0.401 | 67.60 | 0.564 | 82.81 | 0.781 |
| DeepCoral | 82.34 | 0.841 | 86.53 | 0.851 | 66.16 | 0.690 | 62.60 | 0.346 | 72.72 | 0.605 | 74.31 | 0.649 |
| CLUDA | 78.21 | 0.802 | 82.45 | 0.854 | 67.03 | 0.641 | 64.92 | 0.324 | 65.57 | 0.504 | 73.77 | 0.699 |
| SinkDiv | 73.11 | 0.713 | 85.13 | 0.876 | 69.64 | 0.720 | 64.97 | 0.376 | 67.16 | 0.578 | 70.98 | 0.648 |
| Raincoat | 73.11 | 0.713 | 89.13 | 0.873 | 62.11 | 0.603 | 66.22 | 0.357 | 62.11 | 0.523 | 69.09 | 0.727 |
| SSSS-TSA | **99.01** | **0.985** | **90.12** | **0.901** | **72.19** | **0.737** | 66.38 | **0.419** | **75.19** | **0.635** | **83.57** | **0.816** |

## 4 Experiments and Results

In this section, we provide results on multiple realworld timeseries datasets used in prior work, as well as a number of synthetic and corrupted data experiments which test the robustness of our model to different types of domain shift. Finally, we show the benefits of the model in explainability.

### 4.1 Datasets and tasks

**Simulated mean shift data.** We first consider simulated data consisting of 4 dimensional sequences of length 128. Source domain data consists of Gaussian i.i.d. data with variance 1, and the means of these channels shifts between 4 classes. The target domain data is generated by randomly selecting and shifting one channel mean for each class.

**UCI Human Activity Recognition (UCIHAR) .** Data was collected from a group of 30 volunteers performing six different activities (walking, walking upstairs, walking downstairs, sitting, standing, and laying). Each participant was equipped a smartphone with embedded accelerometer and gyroscope that provides 9 channel data (3 axis body acceleration, 3 axis angular velocity, and 3 axis total acceleration)(Anguita et al., 2013).

**Heterogeneity Human Activity Recognition (HHAR).** Data consists of different users carrying various types of smartphones and smartwatches while performing common activities like walking, sitting, standing, etc. This dataset poses a significant challenge in domain adaptation due to the variability in sensor outputs across different devices. This data consists of 3 axis accelerometer data (Stisen et al., 2015).

**PXECG.** This is a 12 channel ECG dataset with 5 diagnostic classes. The data is collected from 5 different sites, each of which constitutes a different domain (Wagner et al., 2020).

**Wireless Sensor Data Mining (WISDM).** Involves time-series data collected from wireless sensors embedded in smartphones. The data typically includes activities like jogging, walking, ascending and descending stairs, sitting, and standing. WISDM can be a highly imbalanced dataset across subjects, which makes it particularly challenging for domain adaptation (Kwapisz et al., 2011).

**WISDM-Bal.** We also take the WISDM dataset and balance classes across source and target domains to better analyze the performance of our proposed method. We do this to ignore issues in domain adaptation that arise because of imbalanced datasets, an issue that we do not aim to address in this paper.

As the possible number of domain adaptation scenarios on WISDM, HHAR, and UCIHAR can be as large as $\binom{30}{2}$, we select 10 domain adaptation scenarios for our experiments. More details on the datasets, the chosen domain adaptation scenarios, and their corresponding results can be found in Appendix D.

## 4.2 Evaluation setup

**Baselines.** The models compared in these experiments include: supervised learning on the source domain (no domain adaptation), Domain-Adversarial Neural Networks (DANN) (Ganin et al., 2016), Adversarial Spectral Kernel Matching (AdvSKM) (Liu & Xue, 2021), CoDATS (Wilson et al., 2020), Domain Adaptation via Sparse Associative Structure Alignment (SASA) (Cai et al., 2021), Conditional Domain Adversarial Networks (CDAN) (Long et al., 2018), DeepCoral (Sun & Saenko, 2016), CLUDA (Ozyurt et al., 2023), Sinkhorn Divergence (SinkDiv) , and Raincoat (He et al., 2023), alongside our proposed method, SSSS-TSA. The Sinkhorn divergence baseline is the Raincoat baseline without the frequency domain information. It aligns the Sinkhorn distance between source and target representations while minimizing the source classification loss. We select these baselines as these are the methods considered in more recent domain adaptation papers (He et al., 2023; Ozyurt et al., 2023). The Adatime benchmarking suite is adapted and used to run these baselines (Ragab et al., 2023).

**Experiments and evaluation details.** We use a 1D CNN neural network as an encoder for all of our baselines. Most of these datasets provide standardized splits to train models and test splits to report numbers. As unlabelled target domain data is not available in real world domain adaptation settings, there is some uncertainty in the community regarding the best way to evaluate domain adaptation methods. We run all methods for a fixed number of epochs and report numbers at the end of these. While less common, this scheme has been used by other papers (e.g., (He et al., 2023)) and most accurately depicts real-world domain adaptation settings. We also report test-set numbers when models attain their best performance on a validation holdout from the training data in the appendix (as that is a popular evaluation criteria in literature). We report both macro F1 and accuracy scores for better evaluation across datasets with varying class imbalances. Each model is run five times on each dataset to ensure statistical reliability, and the results are averaged to produce the mean accuracy and macro F1 scores.

## 4.3 Results on time-series classification benchmarks

The results, as shown in Table 1, demonstrate the superior performance of SSSS-TSA across a range of realworld datasets. We first note that it is interesting to see how poor the performance of most baselines is on the simulated dataset. Our method's performance of 0.98 F1 score underscores how alignment of *channels* can lead to significant improvements over approaches that only align fused or global representations. Our method achieves the highest accuracy and macro F1 score on every dataset considered, except PXECG where the accuracy of our method is lower than the best method by only 0.01%. It often significantly outperforms the second best algorithm, and there is no alternative algorithm that is consistently close to the performance of our approach.

These results underscore the effectiveness of our approach in domain adaptation for time-series data. Our method consistently outperform other state-of-the-art models in various complex and real-world scenarios. The significant improvement in mean macro F1 scores across these datasets highlights the robustness and adaptability of our approach, particularly in handling the challenges posed by multi-channel and noisy time-series data. The Appendix contains additional results when the target labels in validation sets are used to determine the stopping time. While this is not a realistic metric in a true domain adaptation scenario (as it allows labels in the target domain to influence the training process), we find that using this evaluation criteria our method continues to be either competitive or superior to baselines.

## 4.4 Testing across different types of domain shift

To further test the capability of our method to handle different types of shifts, we create additional domain shifts with in the 9-channel UCIHAR dataset. In these experiments, the source is data from one individual (S-2) and the target domain is a new individual (S-11). For the domain adaptation setting, we study three types of additional shifts in the target domain to evaluate the robustness of our approach: (i) *Additive Gaussian noise:* This shift simulates the presence of sensor noise and other measurement artifacts that can degrade the quality of the recorded data. In our experiments, we add Gaussian noise (zero mean, var=2) to randomly selected channels in the target domain that are unobserved in the source domain; (ii) *Saturating channels:* This shift simulates the effect of sensor saturation, where the recorded signal reaches a maximum

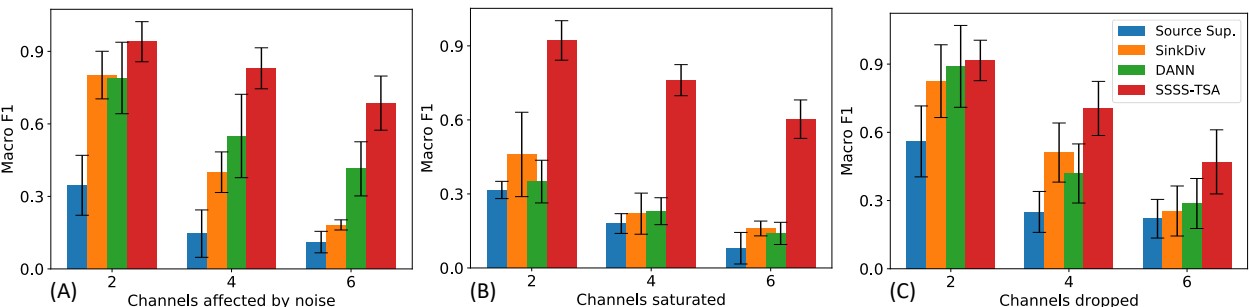

Figure 3: **Domain adaption performance when target domain in UCIHAR dataset is further shifted through corrupted channels.** From left to right, we compare our method to baselines in an (a) additive Gaussian noise setting, (b) saturated channels setting, and (c) when channels are dropped. Our method SSSS-TSA (in red) performs much better.

value and can no longer capture variations in the signal. To simulate channel saturation, we set randomly selected channels to a large value (we set these channels to 2, which is a larger value for these prepossessed normalized datasets). This happens in many microscopy datasets, and sensor networks, when photobleaching occurs or a sensor become faulty; *Dropping channels:* This shift examines the impact of having an unequal number of channels between the source and target domains. To simulate this type of corruption, we drop channels by setting randomly selected channels to 0. By selectively dropping channels in the target domain, we can study the model's ability to adapt when the available information in the target domain is incomplete or differs from the source domain.

For all these cases, we can vary the number of channels (2, 4, 6) affected by these shifts to evaluate the robustness of domain adaptation methods under different levels of channel corruption. This systematic evaluation allows us to understand the resilience and adaptability of our approach in the presence of various types of channel distortions, providing insights into its practical applicability in real-world scenarios.

**Results.**    We repeat our experiment five times, each time using a different random set of selected channels. The average macro F1 scores are shown in Fig. 3. We can see that SSSS-TSA consistently outperforms baselines such as DANN and SinkDiv. We selected these baselines as most other baselines are variants of these. We observe that our baselines exhibit significant performance degradation when only 2 channels are saturated in Fig. 3 (B). Our method achieves a macro F1 score of 0.922 while the best next baseline, SinkDiv Alignment, falls to 0.46. Even when 6 out of the 9 channels are saturated, SSSS-TSA still attains an F1 score of 0.60. Fig. 3 (C) shows the results of the setting where randomly selected channels are dropped. Though our method performs better than other baselines, the margin is smaller for SSSS-TSA when six channels are dropped. As this dataset was preprocessed and normalized, many channels in the source domain would have values closer to 0, which would make it likely for some of these channels to be aligned with dropped channels. Our method performs much better in the additive Gaussian setting as the corrupted channels are less similar to the source channels. Overall, these results show the our method often has the ability to screen and select important unaffected channels to improve time series domain adaptation performance.

### 4.5    Ablations

We conduct multiple ablation studies which investigate how different components of our proposed method affect performance. These ablation results are provided in Table 2.

**How important is it to align channels independently?**    In our first ablation study ( denoted as W/O Ind Align), we evaluate the importance of aligning independent channels in our method. This ablation corresponds to removing the third and fourth terms in equation 2. We can consistently observe across datasets that performance significantly decreases when individual alignment is removed, with performance dropping by 42 % on the UCIHAR dataset. This drop in performance supports our hypothesis on the importance of obtaining informative channel-level representations.

**How important is the classifier for each channel?**    For our second ablation study (W/O Ind Clfr), we investigate how important it is to include channel specific classifiers while aligning representations for

Table 2: **Mean F1 score over 5 runs for different sets of ablations**

|  | W/O Ind Align | W/O Ind Clfr | W/O Attn | W/O Sink | SSSS TSA |
|---|---|---|---|---|---|
| UCIHAR | $0.527 \pm 0.091$ | $0.571 \pm 0.090$ | $0.887 \pm 0.065$ | $0.867 \pm 0.069$ | $\mathbf{0.901} \pm 0.051$ |
| HHAR | $0.399 \pm 0.056$ | $0.503 \pm 0.051$ | $0.717 \pm 0.041$ | $0.604 \pm 0.051$ | $\mathbf{0.737} \pm 0.047$ |
| WISDM | $0.449 \pm 0.094$ | $0.506 \pm 0.054$ | $0.597 \pm 0.046$ | $0.625 \pm 0.049$ | $\mathbf{0.635} \pm 0.053$ |
| WISDM-Bal | $0.639 \pm 0.083$ | $0.645 \pm 0.097$ | $0.694 \pm 0.087$ | $0.781 \pm 0.041$ | $\mathbf{0.816} \pm 0.031$ |
| PKECG | $0.234 \pm 0.026$ | $0.312 \pm 0.031$ | $0.381 \pm 0.040$ | $0.385 \pm 0.046$ | $\mathbf{0.419} \pm 0.031$ |

individual channels. This corresponds to removing the fourth term in equation 2 which results in a setting where Sinkhorn divergence aligns representations for each channel without source class information. This ablation results in a method which is similar to the method proposed by (Chen et al., 2022) in which the authors try to obtain indiscriminate features for different data sensors while not utilizing sensor specific classifiers. We can observe that removing channel specific classifier consistently leads to significantly worse performance, with scores decreasing on UCIHAR dataset by 37 %. This suggests that source class information is critical while aligning individual representations which further supports our intuition provided in 3.3.

**How important is the selection and screening module?** Our third ablation focuses on the impact of the selection and screening module, specifically our implementation of the attention mechanism. When the attention mechanism was removed (W/O Attn), the score decreases across all datasets. The most significant of these decreases was observed on the WISDM-Bal dataset which saw a score reduction of 15%. The attention mechanism enables the model to focus on the most relevant and informative channels, thereby improving the both the quality of the merged representation for classification in the source domain as well as the relevance of this representation in the target domain, and consequently, improving the overall performance of the model.

**What if you change the discrepancy measure?** We also tested our model in an ablation (W/O Sink) were we replace Sinkhorn divergence with the Maximum Mean Discrepancy (MMD) as the discrepancy measure in equation 2. We found that the Sinkhorn outperforms the MMD on all of the datasets we tested, with larger some gaps on some datasets and more modest performance boost on the WISDM dataset. The MMD is sensitive to the choice of bandwidth used in the radial basis kernel. This can potentially cause improved performance in some settings as compared to others. Additionally, the type of shift encountered between source and target domains can vary across datasets, and this can lead to a varying degree of performance gains when utilizing one discrepancy measure over another.

**What is the effect of ablations as more target channels are corrupted?** Of all the ablations that we present in Table 2, the W/O Sink and W/O Attn ablations have the smallest impact on performance. To further investigate this observation, we test W/O Sink and W/O Attn ablations on target channel corruptions that were introduced in section 4.4. We can see in Figure 4 how the performance worsens for the W/O ablation as we increase the number of affected channels. This further supports how our proposed method's channel selection and screening module performs more robustly in the presence of channel level shifts. The W/O Sink ablation, which replaces Sinkhorn divergence with MMD, further supports this observation. The channel selection and screening layer allows W/O Sink ablation to perform better than W/O Attn ablation as more channels are either saturated or dropped.

Interestingly, when channels are affected by noise, the W/O Sink ablation performs worse than the W/O Attn ablation. This observation holds true even as the number of affected channels are increased. These results suggest that for certain distribution shifts, MMD might not be effective at capturing the correct alignment across source and target domains. The Sinkhorn divergence's ability to better capture geometrical properties leads to improved performance across all of these target channel corruption scenarios (Feydy et al., 2019).

We also provide additional ablations in Appendix C.1 where we investigate how performance changes when the same encoder is used for aligning channel level representations. Ablations in Appendix C.2 further explore how employing a vanilla self-attention for the channel selection and screening layer affects performance.

## 4.6 Visualizing the learned weights

To provide further insights into how our channel-selection mechanism works, we visualized the channel weights learned by our model for both source and target domain data (HHAR) in Figure 5. Note that the overall

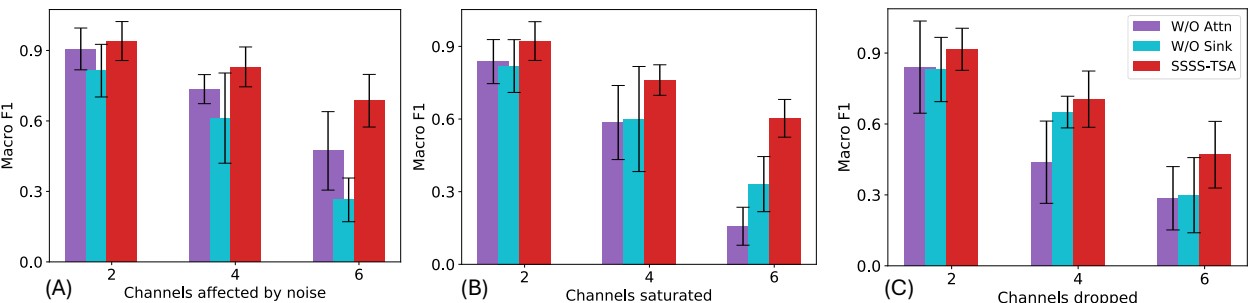

Figure 4: **Ablation study results for the setting where target domain in UCIHAR dataset is further shifted through corrupted channels.** From left to right, we compare our method to baselines in an (a) additive Gaussian noise setting, (b) saturated channels setting, and (c) when channels are dropped.

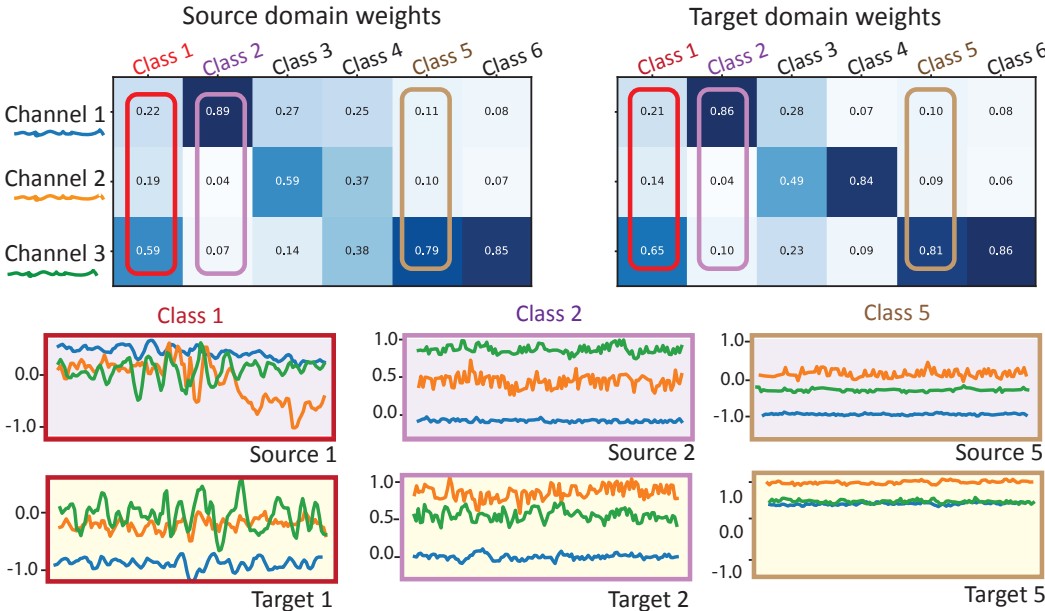

Figure 5: **Visualization of channel selection process.** The top row shows channel weights learned by our model on the HHAR dataset for the corresponding input data below. The overall weight distribution across the two domains is mostly similar. The colored boxes in the matrices highlight how weights learned for different classes help mask channels with larger shifts between the domains, contributing to improved domain adaptation performance.

distribution of the weight matrices across the classes is mostly similar between the source and the target domains. We use source and domain data from three classes to illustrate how these learned weights help improve domain adaptation.

The source and target weights for class 1, bounded in red boxes, select channel 3 as an important channel for the domain adaptation task. We can see in the input data for class 1 that these chosen weights indeed help ignore the blue and the orange channels which encounter major shifts between the source and the target domains. The selected green channel is most similar between the two domains. We observe similar phenomena for classes 2 and 5. While not depicted in the figure, it is also noteworthy that the weights learned for class 4 are quite different in the source and target domains. Upon detailed inspection, each channel is quite informative, and in this case the precise selection of channels is not particularly important for ensuring correct classification.

## 4.7 Visualizing the latent representations

Finally, we also examine the latent representations learned by our models and compare them to a standard Sinkhorn alignment across all channels in Fig. 6. In this example, we can see that the representations formed

by our method provide good overall alignment across all the classes globally and also gives a good local alignment of each class. In contrast, for the Sinkhorn baseline we see that some classes are often fractured and can be mapped to different parts of the latent space.

## 5    Conclusion and Future Directions

A key component of our method is its contrast with traditional encoders that indiscriminately process all channels jointly. Such encoders often fail to exploit the inherent structure and importance of different channels in time-series data, potentially leading to suboptimal performance, especially in the presence of irrelevant or noisy channels. Our approach avoids this pitfall by focusing on the most relevant channels, ensuring that the model is not only more efficient but also more effective in capturing the nuanced relationships within the data. We show how channel invariance can be a powerful tool for alignment, especially in noisy channel settings and across different subsets of channels in the training and testing sets.

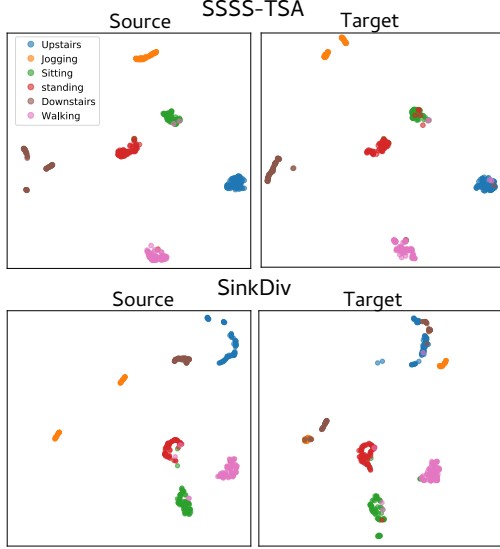

Figure 6: **Visualization of latents for our method and baseline.** Umap embeddings for domain adaptation across two subjects on the HHAR dataset for (Top) our approach and the (Bottom) Sinkhorn Divergence baseline.

Of course our approach is not without limitations. One such limitation is the potential for overfitting in scenarios with extremely noisy or sparse datasets, where channel selection might become biased towards non-representative features. Additionally, our current model uses different encoders for each channel which restricts the application to new or unknown channels. Nevertheless, our method has shown the advantages of selectively screening and aligning channels representations. This can likely be used in conjunction with other recent methods that explore augmentations (Ozyurt et al., 2023) and label correction (He et al., 2023) for further improvements in the future.

There is further scope to extend our method to multi-modal domain adaptation settings. In such settings, some modalities provide more information in aligning the correct source and target representations. Our method can be adapted for such scenarios where selective alignment of individual modalities could lead to improved domain adaptation performance. Examples of such multi-modal settings could potentially involve temporal and frequency modal information for time series, as well as more generalized multi-model settings involving audio and video modalities.

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

# Appendix

## A    Details for Sinkhorn divergence

As mentioned earlier in the main paper, Sinkhorn divergences are variants of Wasserstein distances that compute an entropic regularized optimal transport plan between two sets of distributions.

To transport between two sets of points, can express these distributions as $\boldsymbol{\alpha} = \sum_{i=1}^{n} a_i \boldsymbol{\delta}_{\boldsymbol{x}_i}$ and $\boldsymbol{\beta} = \sum_{j=1}^{m} b_j \boldsymbol{\delta}_{\boldsymbol{y}_j}$, where $\boldsymbol{\delta}_{\boldsymbol{x}}$ is the Dirac function at position $\boldsymbol{x} \in \mathbb{R}^d$, so that the $\boldsymbol{x}_i$ and $\boldsymbol{y}_j$ denote the mass locations for the distributions and $a_i, b_i \in \mathbb{R}_+$ are the weights at these mass locations for $\boldsymbol{\alpha}$ and $\boldsymbol{\beta}$ respectively. The Sinkhorn distance between two distributions $\boldsymbol{\alpha}$ and $\boldsymbol{\beta}$ is defined as

$$\mathcal{S}_\gamma(\boldsymbol{\alpha}, \boldsymbol{\beta}) = \min_{\boldsymbol{P}} \ \langle \boldsymbol{C}, \boldsymbol{P} \rangle - \gamma \boldsymbol{H}(\boldsymbol{P}), \ \ \text{s.t} \ \boldsymbol{P} \in \mathbb{R}_+^{n \times m}, \boldsymbol{P}^T \mathbb{1}_n = \boldsymbol{b}, \boldsymbol{P} \mathbb{1}_m = \boldsymbol{a}, \tag{3}$$

where $\boldsymbol{P}$ is called the transport plan, the ground cost metric $\boldsymbol{C} \in \mathbb{R}^{n \times m}$ represents the transportation cost between each pair of distribution mass locations, $\boldsymbol{H}(\boldsymbol{P})$ is the entropy of the transport plan matrix $\boldsymbol{P}$ and is given by $\boldsymbol{H}(\boldsymbol{P}) = \sum_{i=1}^{n} \sum_{j=1}^{m} \boldsymbol{P}_{i.j}(\log \boldsymbol{P}_{i,j} - 1)$, while $\gamma$ is a regularization parameter. This regularization term makes the minimization problem strongly convex and makes it less sensitive to changes in input, and can be solved with $O(n^2)$ computations using the Sinkhorn algorithm Cuturi (2013).

## B    Hyperparameters and other training details

For all of our runs, we used a Sinkhorn regularization parameter, $\gamma = 1e - 3$.
We used the ADAM Optimizer with a learning rate set to $1e - 3$ for all experiments. Our loss function doesn't take a constant sum of the supervised classification loss and Sinkhorn alignment values. The ratio of these two terms in the loss function wasn't tuned for any experiment. We were mindful of how our experiments settings should reflect real world scenarios where true labels from the target domain aren't available. All datasets were trained for 300 epochs before reporting numbers in table 1 For the HHAR and WISDM datasets, the temperature parameter $\tau$ for the softmax non linearity was set to 3. For UCIHAR this was set to 9 (as a larger number of channels were involved). For the remaining baselines, we used the already set hyperparameters for these different datasets in the Adatime benchmarking suite. We used Raincoat's adaptation of this benchmarking suite https://github.com/mims-harvard/Raincoat to run both raincoat and other baselines. All experiments were performed on a Single NVIDIA Quadro RTX 5000. More results on hyperparamaters and their sensitivity can be found in C.6.

### B.1    Training, validation and testing splits.

The publicly available datasets we report numbers on already contain train and test splits for each domain adaptation scenario(which are also used by the Adatime benchmarking suite). We use the same splits as Adatime. For results, such as those in Table A3, that require a held out validation set, we split the dedicated training set in these benchmarking datasets into a random %70/%30 split. The larger split was used for training and the smaller split was used for validation. Results were reported on the pre-designated test splits provided by these benchmarks.

## C    Additional Results

### C.1    Using a single encoder for aligning representations across all channels

A key ingredient behind our approach is the use of channel level representations. We tested whether having different weights for each channel improves performance. We provide these results in Table A1).Our results demonstrate that having different weights for our CNN encoder across channels provides large gains, with a shared encoder yielding a performance of 0.67 and our method with different channel encoder weights gives a score of 0.82.

Table A1: *Ablations when a single encoder is used for aligning all channels*

| Dataset | Shared | Ours |
|---|---|---|
| UCI-HAR | 0.65 | 0.90 |
| HHAR | 0.58 | 0.74 |
| WISDM | 0.52 | 0.64 |
| WISDM-Bal | 0.67 | 0.82 |

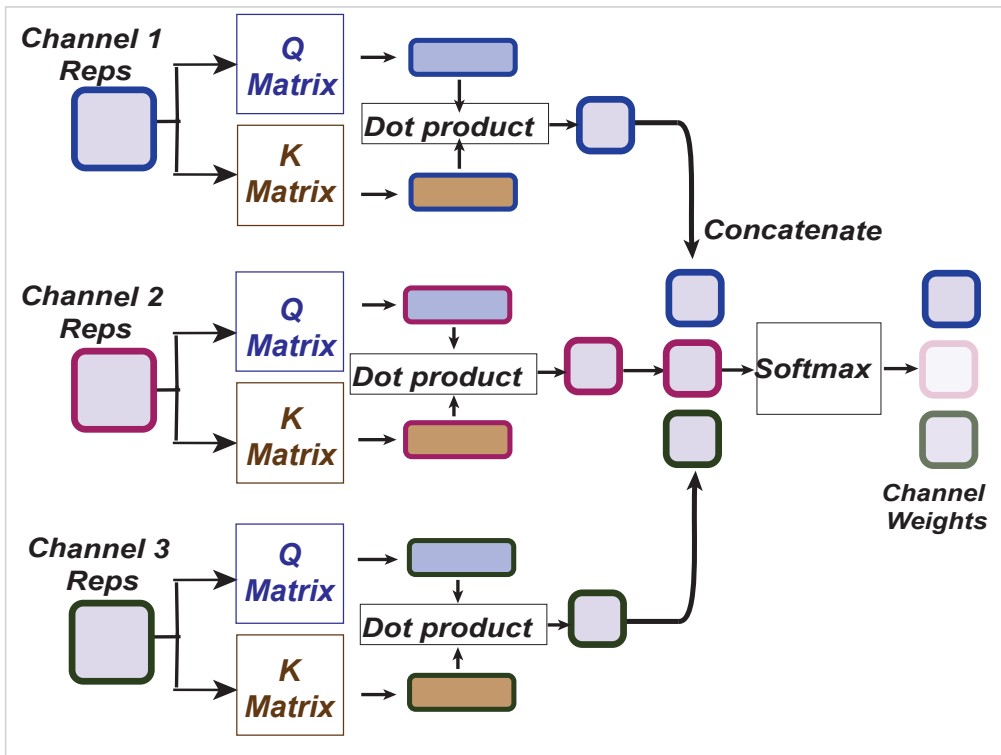

Figure A1: Signal Selection Layer in Figure 2 for source domain.

## C.2 Adding
**in a more complex selection procedure with self-attention**

In domain adaptation alignment settings, the weight for a channel
should depend only on the information contained within the channel; information from other channels could potentially corrupt this weight computation. To confirm this hypothesis, we ran experiments where we replaced our channel selection mechanism with a vanilla self-attention layer for computing weights $\boldsymbol{w}$ in 1. A vanilla self-attention layer would compute a channel weight matrix that depends on the interaction between different channels. Ablation results in Table A2 confirm our hypothesis as we can see that the performance degrades when vanilla self-attention is used. Our channel selection operation can also be expressed as a restricted self attention layer that doesn't attend to cross channel attention weights:

$$\boldsymbol{w} = \text{Softmax}\left(\text{diag}\left(\frac{\boldsymbol{QK}^{\intercal}}{\sqrt{d}}\right)\right)\mathbf{I},$$

where the *values* matrix used in vanilla self attention is replaced by the identity matrix $\mathbf{I}$, and only the diagonal terms of the self-attention matrix are retained.

## C.3 Learned weights for corrupted channels

We also obtained results in figure A2 that show the distribution of the learned weights by the corrupted channel in experiments of 3.

We computed these statistics by obtaining the fraction of total number of channels that were either dropped, saturated or added with noise, and obtained their corresponding weights. These diagrams are histograms with points plotted at the midway point of the histogram bins. Multiple could occlude each other, so we show line plots with markers at the mid-point of the histogram bins.

Table A2: *Ablations comparing vanilla self-attention with our proposed attention method.*

| Dataset | Self-Attn | Ours |
|---|---|---|
| UCI-HAR | 0.85 | 0.90 |
| HHAR | 0.66 | 0.74 |
| WISDM | 0.60 | 0.64 |
| WISDM-Bal | 0.75 | 0.82 |

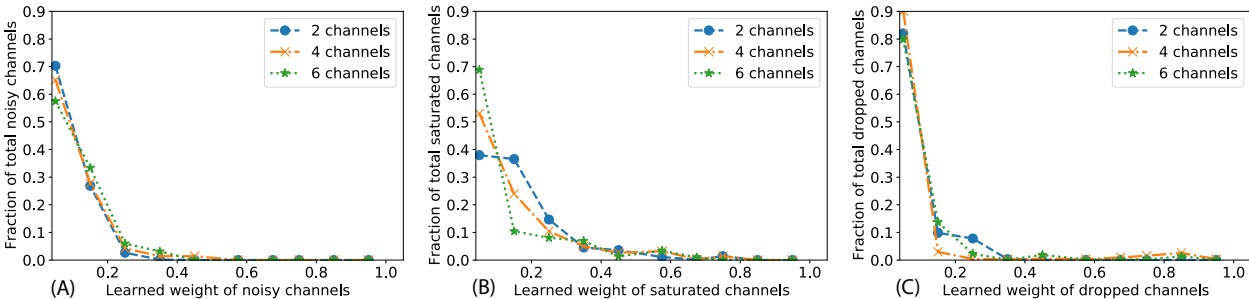

Figure A2: **Learned weights for corrupted channels.** For the experiments in figure 3, we show the fraction of corrupted channel and their corresponding learned weights.

## C.4 Different testing schemes

Table A3 shows macro F1 scores when target domain labels in the
validation hold out set were used to report evaluation numbers. The model parameters corresponding to the best macro F1 performance on the target domain holdout validation set were saved. The performance of these saved models on test sets was then reported. We believe this method doesn't accurately reflect real world performance, but report numbers here as this is a common evaluation scheme used in literature.

Table A3: Mean macro F1 scores over 5 runs for different domain adaptation methods

| Model | Simulations | UCIHAR | HHAR | WISDM | WISDM-Bal | PXECG |
|---|---|---|---|---|---|---|
| Supervised Src | 0.262 | 0.836 | 0.66 | 0.504 | 0.521 | 0.366 |
| DANN | 0.7 | 0.891 | 0.701 | 0.681 | 0.626 | 0.361 |
| AdvSKM | 0.712 | 0.88 | 0.671 | 0.616 | 0.665 | 0.389 |
| CoDATS | 0.531 | 0.907 | 0.744 | **0.685** | 0.816 | 0.366 |
| CDAN | 0.812 | 0.647 | 0.731 | 0.632 | 0.742 | 0.363 |
| SASA | 0.654 | 0.803 | 0.681 | 0.559 | 0.801 | 0.396 |
| DeepCoral | 0.843 | 0.892 | 0.697 | 0.621 | 0.701 | 0.346 |
| CLUDA | 0.802 | 0.857 | 0.661 | 0.491 | 0.760 | 0.325 |
| SinkDiv | 0.713 | 0.876 | 0.720 | 0.602 | 0.648 | 0.376 |
| Raincoat | 0.713 | 0.889 | 0.714 | 0.519 | 0.746 | 0.354 |
| SSSS-TSA | **0.98** | **0.915** | **0.787** | 0.677 | **0.857** | **0.422** |

It can be seen that there is a big difference in methods that employ adversarial learning (such as DANN, CoDATS etc.) between numbers in Table A3 and 1. Methods employing Sinkhorn Distance have a smaller margin difference between these two approaches. Adversarial methods can be very unstable, and the best target domain validation scores often do not correspond to higher source domain classification scores. Source domain F1 scores on validation holdout sets are another way to stop model training and report domain adaptation methods on target domain test sets, but source classification scores reach their best values very early on in the training regime, way before target domain performance improves to their best values.

For HHAR and UCIHAR datasets, we use the same domain adaptation scenarios used in He et al. (2023). For WISDM we use a different set of domain adaptation scenarios as the raincoat paper addresses the universal domain shift problem, where some source domain calsses are not present in the target domain. As that is not the focus of our paper, we use a different set of domain adaptation scenarios for the WISDM dataset.

## C.5 Computational complexity

As our method employs separate encoder and classifier for each channel, its important to analyze the computational and memory demands to better understand scaling ability. We observe that time series

encoders such as 1D CNN are much smaller in size than vision or language encoders and this results in manageable memory demands. For all our experiments, we used an encoder that consisted of 3 layers of 1D CNN. The size of each of these channel-specific encoders was 0.38014 MB. Each channel-specific classifier was of size 0.065MB. Such a small memory footprint would allow our model to adequately handle memory usage for high-dimensional input time series datasets. These results imply that SSSS-TSA would take less than 50 MBs of memory when used on a 100 channel dataset.

These channel-specific encoders also have manageable computational complexity. For the 9 channel UCI HAR dataset, the time taken for a batch of 64 samples in the forward pass is 0.04700 seconds (which involves passing the source and target data through channel-specific encoders, then channel-specific classifiers, then obtaining channel-specific alignment, pooling representations through the attention layer, and then finally align and classifying globally pooled representations). The backward pass takes 0.0363 seconds and updating the weights of the model takes 0.02573 seconds. This totals to 0.109 seconds.

For comparison, a scheme where no channel-specific encoders and classifiers are used (SinkDiv alignment), the total time for forward, backward passes and weight updates for one batch is 0.015244 seconds.

This shows that when separate encoders, classifiers, and alignment plans are used for each channel, the time complexity scales by slightly less than the number of channels in input. Thus for a 100 channel time series dataset, the total update in a training iteration would take less than 1.5 seconds.

Our implementation was done on a single NVIDIA Quadro RTX 5000 GPU and consisted of iterating over all channels encoders, classifiers, and transport plans sequentially. For higher dimensional time series, multiple GPUs and multi-threading for different encoders can be used to further speed up computations and improve time complexity.

### C.6 Hyperparameter sensitivity and tuning

We provide results below which analyze the sensitivity of our method to different hyperparamaters. All results below are reported on trained models that achieve best performance on a holdout validation target set.

#### C.6.1 Entropic regularization weight

The entropic regularization weight of the optimal transport plan, $\gamma$, is the main hyperparameter involved in computing Sinkhorn divergences. As $\gamma$ increases, the optimal transport plan smooths/equalizes. This implies that all points in the source domain are transported to all points in the target domain. To further investigate the effect of $\gamma$, we analyze how results change when $\gamma$ is varied on the 3 channel WISDM-Bal and 9 channel UCIHAR datasets. We can see in Table A4 that F1 scores are within a range of $\pm 0.05$ when the $\gamma$ parameter is between 1e-1 and 1e-4. For a gamma value greater than and equal to 1, there is a sharp drop across both datasets. This is expected for larger values of $\gamma$ as the resulting transport plan can cause Sinkhorn

Table A4: *Sensitivity to entropic regularization parameter $\gamma$.*

| $\gamma$ | WISDM-Bal | UCIHAR |
|---|---|---|
| 1e-4 | 0.8801 | 0.914 |
| 1e-3 | 0.857 | 0.915 |
| 1e-2 | 0.864 | 0.889 |
| 1e-1 | 0.915 | 0.883 |
| 1 | 0.708 | 0.781 |
| 10 | 0.691 | 0.701 |

divergence to lose its geometric discerning properties which can affect performance. As target labels are not available for domain adaptation problems, we do not tune the gamma parameter. We set it to 1e-3 for all experiments. Other works in literature such as (He et al., 2023), also set $\gamma$ to 1e-3 . Another reason for setting $\gamma = 1e-3$ is that it is shown to provide a balance between performance and computational efficiency which we further discuss below.

### C.6.2   Number of iterations for Sinkhorn divergence

The number of Sinkhorn iterations required for computing Sinkhorn divergence depends on the set iteration tolerance $\epsilon$ for Sinkhorn potentials. Table A5 shows the number of iterations required for different values of $\gamma$ at different values of $\epsilon$. For $\epsilon = $ 1e-3, we can see that the number of iterations is very large when $\gamma$ values are small. One commonly used approach to speed up Sinkhorn divergence with minimal loss in performance, is to relax the $\epsilon$ tolerance. This has been explored in literature where the authors show on simulated data how only 5 iterations at smaller values of $\gamma$ can provide a good balance between performance and computation speed (Bigot et al., 2022) . An $\epsilon$ value of 1e-1 was also used by other domain adaptation papers (He et al., 2023) . For these reasons, we used an epsilon value of 1e-1 for all our experiments in our paper (including Table 1 results). A value of 1e-1

Table A5: *Number of Sinkhorn iterations required at iteration tolerance values $\epsilon$ for different values values of $\gamma$.*

| $\gamma$ | $\epsilon = 0.001$ | $\epsilon = 0.1$ |
|---|---|---|
| 1e-4 | 734 | 5 |
| 1e-3 | 494 | 5 |
| 1e-2 | 274 | 5 |
| 1e-1 | 59 | 4 |
| 1 | 8 | 3 |
| 10 | 5 | 3 |

results in a lower number of required iterations. This provides us the right balance between computation efficiency and performance.

Fp the WISDM-Bal dataset which consists of a batch of size 64, where each channel encoded embeddings had a size of 64, the time needed to compute Sinkhorn divergence between source and target channel representations was 0.00135 seconds at epsilon value of 1e-1 and $\gamma$ value of 1e-3.

A more theoretical convergence analysis is perhaps beyond the scope of this paper, but there are many useful works in the literature that provides such analysis (Ghosal & Nutz, 2022).

### C.6.3   Temperature parameter for Channel selection and screening layer

We also analyze the sensitivity of the channel selection and screening layer temperature parameter $\tau$. We analyze the effect of varying $\tau$ through experiments on the 3 channel WISDM-Bal dataset and the 9 channel UCIHAR dataset. For both of these datasets, we selected domain adaptation settings that saw SSSS-TSA make large gains over non channel selection baselines such as SinkDiv. This helped us identify cases where the channel selection screening layer is critical for performance improvement, thus helping us better identify how changes in $\tau$ affect the behavior of the channel selection layer. For the WISDM-Bal dataset, we analyzed the source 20 to target 30 domain adaptation scenario. This scenario had a 0.6638 F1 score on the SinkDiv Alignment baseleine. For the UCIHAR dataset, we analyzed the source 12 to target 16 domain adaptation scenario.

This scenario had a 0.6662 F1 score on the SinkDiv Alignment baseline. We can see that the results are mostly similar across different values of $\tau$, except for $\tau = 0.1$ on the 9 channel UCIHAR dataset. For smaller values of $\tau$, the channel selection weights provided by the softmax distribution are encouraged to have a low entropy. This results in much larger weights to be assigned to a few channels, while most other channels are ignored. As with other hyper-parameters, target labels are often not available in real-world domain adaptation settings. For this reason we set a rule for selecting the parameter. equal to the number of channels, so that entropy of the channel selection scheme scales with the number of input channels. Though the correct temperature value would depend on the channel informativeness/ and use case practitioners are interested. If practitioners think that almost

Table A6: *Sensitivity to selection and screening layer temperature parameter $\tau$.*

| $\tau$ | WISDM-Bal ($20 \rightarrow 30$) | UCIHAR ($12 \rightarrow 16$) |
|---|---|---|
| 0.1 | 0.884 | 0.807 |
| 1 | 0.896 | 0.858 |
| 3 | 0.891 | 0.851 |
| 5 | 0.898 | 0.855 |
| 10 | 0.898 | 0.862 |
| 20 | 0.897 | 0.871 |
| 50 | 0.878 | 0.852 |

all channels should be ignored and very few cahnnels should be selected (and many channels are noisy), then a small value of would be suitable. If practitioners think only few channels should be dropped then a larger value of $\tau$ would be preferable. Classification performance on source domain validation held-out labels could be a good indicator for determining which of this is more desirable.

# D  Results for different domain adaptation scenarios

## D.1  WISM domain adaptation scenarios

Table A7: *WISDM scenario test scores at end of training. Mean macro F1 scores for each domain adaptation scenario over 5 runs*

| Case | Supervised | DANN | AdvSKM | CoDATS | CDAN | CLUDA | SinkDiv | Raincoat | SSSS-TSA |
|---|---|---|---|---|---|---|---|---|---|
| 20 to 30 | 0.5545 | 0.5668 | 0.6142 | 0.6144 | 0.4954 | 0.4862 | 0.7039 | 0.2414 | 0.6116 |
| 12 to 19 | 0.3668 | 0.2901 | 0.3380 | 0.2427 | 0.2758 | 0.2715 | 0.3153 | 0.2636 | 0.4451 |
| 30 to 20 | 0.6525 | 0.6540 | 0.6630 | 0.7912 | 0.7156 | 0.3591 | 0.4941 | 0.4506 | 0.7961 |
| 2 to 32 | 0.5897 | 0.4067 | 0.6914 | 0.5667 | 0.5004 | 0.3940 | 0.5003 | 0.3392 | 0.6345 |
| 7 to 30 | 0.7673 | 0.7471 | 0.8101 | 0.6666 | 0.7329 | 0.5073 | 0.6329 | 0.5073 | 0.8932 |
| 12 to 7 | 0.5132 | 0.5467 | 0.5200 | 0.5674 | 0.4905 | 0.3755 | 0.6804 | 0.4070 | 0.4785 |
| 18 to 20 | 0.5604 | 0.5212 | 0.5855 | 0.6886 | 0.5812 | 0.4400 | 0.6477 | 0.3912 | 0.6258 |
| 19 to 30 | 0.5627 | 0.3386 | 0.5657 | 0.3860 | 0.4865 | 0.6343 | 0.3454 | 0.4131 | 0.5105 |
| 4 to 19 | 0.1971 | 0.3404 | 0.3725 | 0.2932 | 0.3196 | 0.4181 | 0.6344 | 0.3119 | 0.6313 |
| 26 to 2 | 0.6626 | 0.6625 | 0.6492 | 0.6672 | 0.5730 | 0.3222 | 0.6162 | 0.2797 | 0.7321 |

Table A8: *WISDM scenario test scores when validation target domain labels to stop early. Mean macro F1 scores over 5 runs*

| Case | Supervised | DANN | AdvSKM | CoDATS | CDAN | CLUDA | SinkDiv | Raincoat | SSSS-TSA |
|---|---|---|---|---|---|---|---|---|---|
| 20 to 30 | 0.6088 | 0.6303 | 0.6227 | 0.6980 | 0.6112 | 0.4862 | 0.6701 | 0.5716 | 0.7303 |
| 12 to 19 | 0.3661 | 0.3884 | 0.3306 | 0.3687 | 0.3247 | 0.2715 | 0.3800 | 0.2904 | 0.4479 |
| 30 to 20 | 0.7447 | 0.8452 | 0.7678 | 0.7959 | 0.8117 | 0.3591 | 0.5757 | 0.5818 | 0.8220 |
| 2 to 32 | 0.6810 | 0.7153 | 0.7160 | 0.7392 | 0.6914 | 0.3940 | 0.5660 | 0.6019 | 0.6295 |
| 7 to 30 | 0.7801 | 0.8041 | 0.8326 | 0.8528 | 0.7985 | 0.5073 | 0.6907 | 0.7195 | 0.9406 |
| 12 to 7 | 0.5335 | 0.7322 | 0.5245 | 0.7215 | 0.6271 | 0.3755 | 0.6925 | 0.5385 | 0.5154 |
| 18 to 20 | 0.6014 | 0.7349 | 0.6403 | 0.7316 | 0.6879 | 0.4400 | 0.7145 | 0.5469 | 0.6621 |
| 19 to 30 | 0.6847 | 0.6636 | 0.5898 | 0.5889 | 0.6716 | 0.6343 | 0.5201 | 0.5136 | 0.5792 |
| 4 to 19 | 0.3741 | 0.5121 | 0.4780 | 0.4719 | 0.4212 | 0.4181 | 0.5868 | 0.4243 | 0.6326 |
| 26 to 2 | 0.6716 | 0.7616 | 0.6627 | 0.8833 | 0.6743 | 0.3222 | 0.6311 | 0.4040 | 0.8181 |

## D.2  WISDM-Balanced

Table A9: *WISDM-balance scenario test scores at end of training. Mean macro F1 scores for each domain adaptation scenario over 5 runs*

| Case | Supervised | DANN | AdvSKM | CoDATS | CDAN | CLUDA | SinkDiv | Raincoat | SSSS-TSA |
|---|---|---|---|---|---|---|---|---|---|
| 20 to 30 | 0.5667 | 0.6495 | 0.6239 | 0.6067 | 0.5970 | 0.6230 | 0.6638 | 0.3382 | 0.8855 |
| 12 to 19 | 0.3898 | 0.6230 | 0.4848 | 0.8289 | 0.6673 | 0.5816 | 0.6306 | 0.5004 | 0.8333 |
| 30 to 20 | 0.5412 | 0.7388 | 0.6059 | 0.7048 | 0.7660 | 0.7410 | 0.6126 | 0.4495 | 0.7644 |
| 2 to 32 | 0.6133 | 0.7250 | 0.7004 | 0.6928 | 0.6007 | 0.6538 | 0.5506 | 0.4664 | 0.7205 |
| 7 to 30 | 0.7890 | 0.7526 | 0.7600 | 0.6975 | 0.7524 | 0.9223 | 0.6591 | 0.3515 | 0.9667 |
| 12 to 7 | 0.6028 | 0.7442 | 0.5418 | 0.5971 | 0.6109 | 0.7500 | 0.7541 | 0.5294 | 0.8966 |
| 18 to 20 | 0.5671 | 0.4784 | 0.6178 | 0.5805 | 0.4810 | 0.6663 | 0.7239 | 0.6197 | 0.5801 |
| 19 to 30 | 0.5572 | 0.6440 | 0.6824 | 0.5357 | 0.7061 | 0.6586 | 0.6913 | 0.3526 | 0.7218 |
| 4 to 19 | 0.1911 | 0.7800 | 0.4673 | 0.7174 | 0.6834 | 0.8079 | 0.7589 | 0.5225 | 0.9284 |
| 26 to 2 | 0.6456 | 0.6997 | 0.6281 | 0.6936 | 0.7472 | 0.5836 | 0.5529 | 0.3287 | 0.8598 |

Table A10: *WISDM-balance scenario test scores when validation target domain labels used to stop early. Mean macro F1 scores over 5 runs*

| Case | Supervised | DANN | AdvSKM | CoDATS | CDAN | CLUDA | SinkDiv | Raincoat | SSSS-TSA |
|------|-----------|------|--------|--------|------|-------|---------|----------|----------|
| 20 to 30 | 0.6094 | 0.6610 | 0.6204 | 0.7028 | 0.6752 | 0.6783 | 0.6586 | 0.7077 | 0.8904 |
| 12 to 19 | 0.3969 | 0.7043 | 0.4847 | 0.9336 | 0.8137 | 0.6398 | 0.6889 | 0.6532 | 0.8765 |
| 30 to 20 | 0.6752 | 0.8003 | 0.7157 | 0.8089 | 0.8301 | 0.7684 | 0.6623 | 0.7803 | 0.7048 |
| 2 to 32 | 0.6761 | 0.8401 | 0.7600 | 0.8243 | 0.7414 | 0.7519 | 0.6099 | 0.7106 | 0.7316 |
| 7 to 30 | 0.7767 | 0.7578 | 0.7628 | 0.7576 | 0.7346 | 0.8588 | 0.7167 | 0.7903 | 0.9830 |
| 12 to 7 | 0.6375 | 0.7893 | 0.6232 | 0.8190 | 0.6694 | 0.8709 | 0.7723 | 0.8969 | 0.9695 |
| 18 to 20 | 0.6402 | 0.6508 | 0.6591 | 0.7890 | 0.6001 | 0.7542 | 0.7116 | 0.7820 | 0.7508 |
| 19 to 30 | 0.7005 | 0.7950 | 0.7336 | 0.8048 | 0.7834 | 0.7585 | 0.7170 | 0.8508 | 0.8642 |
| 4 to 19 | 0.5702 | 0.8186 | 0.6876 | 0.8425 | 0.7823 | 0.8247 | 0.7878 | 0.5886 | 0.9529 |
| 26 to 2 | 0.6621 | 0.7455 | 0.6111 | 0.8788 | 0.7979 | 0.6944 | 0.5934 | 0.6992 | 0.8528 |

## D.3    HHAR domain adaptation scenarions

Table A11: *HHAR scenario test scores at end of training. Mean macro F1 scores for each domain adaptation scenario over 5 runs*

| Case | Supervised | DANN | AdvSKM | CoDATS | CDAN | CLUDA | SinkDiv | Raincoat | SSSS-TSA |
|------|-----------|------|--------|--------|------|-------|---------|----------|----------|
| 0 to 2 | 0.6103 | 0.6879 | 0.6611 | 0.6550 | 0.7097 | 0.6975 | 0.6654 | 0.7291 | 0.8423 |
| 1 to 6 | 0.8185 | 0.9482 | 0.8703 | 0.9296 | 0.9402 | 0.8505 | 0.9193 | 0.9010 | 0.9157 |
| 2 to 4 | 0.3948 | 0.6147 | 0.4327 | 0.5501 | 0.6009 | 0.6029 | 0.6763 | 0.3900 | 0.6581 |
| 4 to 0 | 0.2688 | 0.2749 | 0.2585 | 0.3431 | 0.3104 | 0.3241 | 0.3487 | 0.2324 | 0.4575 |
| 4 to 5 | 0.8245 | 0.9564 | 0.9283 | 0.9633 | 0.9558 | 0.8973 | 0.8914 | 0.9069 | 0.9406 |
| 5 to 1 | 0.9011 | 0.9789 | 0.9000 | 0.9695 | 0.9557 | 0.9335 | 0.9457 | 0.8873 | 0.9794 |
| 5 to 2 | 0.2905 | 0.3671 | 0.3435 | 0.3112 | 0.4011 | 0.4939 | 0.4076 | 0.2422 | 0.5563 |
| 7 to 2 | 0.3604 | 0.4264 | 0.3817 | 0.2890 | 0.4376 | 0.4490 | 0.4351 | 0.3776 | 0.5978 |
| 7 to 5 | 0.6025 | 0.8752 | 0.6194 | 0.8606 | 0.6830 | 0.6075 | 0.7413 | 0.7186 | 0.7142 |
| 8 to 4 | 0.7085 | 0.9727 | 0.7661 | 0.9679 | 0.9714 | 0.5585 | 0.7858 | 0.6518 | 0.6459 |

Table A12: *HHAR scenario test scores when validation target domain labels used to stop early. Mean macro F1 scores over 5 runs*

| Case | Supervised | DANN | AdvSKM | CoDATS | CDAN | CLUDA | SinkDiv | Raincoat | SSSS-TSA |
|------|-----------|------|--------|--------|------|-------|---------|----------|----------|
| 0 to 2 | 0.6971 | 0.7520 | 0.7107 | 0.7207 | 0.7774 | 0.7118 | 0.7285 | 0.7436 | 0.8923 |
| 1 to 6 | 0.9022 | 0.9541 | 0.8986 | 0.9414 | 0.9475 | 0.8739 | 0.9372 | 0.9198 | 0.9344 |
| 2 to 4 | 0.4846 | 0.6613 | 0.4835 | 0.6119 | 0.6264 | 0.6288 | 0.8093 | 0.5962 | 0.7558 |
| 4 to 0 | 0.3090 | 0.4000 | 0.3181 | 0.4166 | 0.3723 | 0.3991 | 0.3683 | 0.4408 | 0.5506 |
| 4 to 5 | 0.8819 | 0.9709 | 0.9445 | 0.9747 | 0.9551 | 0.9064 | 0.9254 | 0.9182 | 0.9382 |
| 5 to 1 | 0.9248 | 0.9776 | 0.9224 | 0.9766 | 0.9599 | 0.9176 | 0.9470 | 0.9472 | 0.9771 |
| 5 to 2 | 0.3699 | 0.5157 | 0.3874 | 0.4631 | 0.4503 | 0.5153 | 0.4521 | 0.4527 | 0.6217 |
| 7 to 2 | 0.4074 | 0.5232 | 0.4078 | 0.4525 | 0.4668 | 0.4744 | 0.4378 | 0.5047 | 0.6245 |
| 7 to 5 | 0.7230 | 0.9008 | 0.7561 | 0.9077 | 0.7767 | 0.6334 | 0.7688 | 0.9253 | 0.8084 |
| 8 to 4 | 0.8425 | 0.9739 | 0.8777 | 0.9757 | 0.9783 | 0.5552 | 0.8279 | 0.6933 | 0.7706 |

## D.4 UCIHAR domain adaptation scenarions

Table A13: *UCIHAR scenario test scores at end of training. Mean macro F1 scores for each domain adaptation scenario over 5 runs*

| Case | Supervised | DANN | AdvSKM | CoDATS | CDAN | CLUDA | SinkDiv | Raincoat | SSSS-TSA |
|------|-----------|------|--------|--------|------|-------|---------|----------|----------|
| 2 to 11 | 0.5842 | 0.9653 | 0.9967 | 0.9509 | 0.8339 | 0.9401 | 0.9732 | 0.9741 | 0.9768 |
| 6 to 23 | 0.7358 | 0.9076 | 0.8739 | 0.9722 | 0.9465 | 0.8969 | 0.8984 | 0.8812 | 0.9212 |
| 7 to 13 | 0.7897 | 0.8758 | 0.8469 | 0.8678 | 0.9025 | 0.8497 | 0.9264 | 0.9123 | 0.9407 |
| 9 to 18 | 0.3897 | 0.5603 | 0.5786 | 0.6434 | 0.5345 | 0.5156 | 0.5837 | 0.5921 | 0.6938 |
| 12 to 16 | 0.4853 | 0.4907 | 0.6208 | 0.5418 | 0.5660 | 0.6082 | 0.6662 | 0.6733 | 0.8155 |
| 13 to 19 | 0.9155 | 0.7635 | 0.9297 | 0.8883 | 0.9426 | 0.9336 | 0.9685 | 0.9540 | 0.9283 |
| 18 to 21 | 0.9947 | 0.9531 | 0.9967 | 0.9939 | 1.0000 | 0.9175 | 0.9966 | 0.9990 | 1.0000 |
| 20 to 6 | 0.9694 | 0.9446 | 1.0000 | 0.9552 | 0.8159 | 0.8321 | 1.0000 | 0.9578 | 0.9463 |
| 23 to 13 | 0.7981 | 0.7455 | 0.7867 | 0.7464 | 0.8379 | 0.7543 | 0.8003 | 0.8732 | 0.8228 |
| 24 to 12 | 0.8380 | 0.9810 | 0.8535 | 0.9967 | 0.9810 | 0.9944 | 0.9555 | 0.9532 | 0.9669 |

Table A14: *UCIHAR scenario test scores when validation target domain labels used to stop early. Mean macro F1 scores over 5 runs*

| Case | Supervised | DANN | AdvSKM | CoDATS | CDAN | CLUDA | SinkDiv | Raincoat | SSSS-TSA |
|------|-----------|------|--------|--------|------|-------|---------|----------|----------|
| 2 to 11 | 0.7290 | 0.9967 | 0.9967 | 0.9935 | 0.9935 | 0.9843 | 1.0000 | 1.0000 | 1.0000 |
| 6 to 23 | 0.7326 | 0.9772 | 0.8800 | 0.9793 | 0.9948 | 0.9252 | 0.9219 | 0.9224 | 0.9742 |
| 7 to 13 | 0.8753 | 0.9232 | 0.8524 | 0.8748 | 0.9556 | 0.8377 | 0.8994 | 0.8610 | 0.9209 |
| 9 to 18 | 0.7231 | 0.6750 | 0.6695 | 0.7402 | 0.7152 | 0.5045 | 0.5634 | 0.6544 | 0.6521 |
| 12 to 16 | 0.6534 | 0.5922 | 0.6351 | 0.6743 | 0.6624 | 0.6131 | 0.6560 | 0.6754 | 0.8571 |
| 13 to 19 | 0.9468 | 0.9758 | 0.9514 | 0.9921 | 0.9654 | 0.9212 | 0.9740 | 0.9547 | 0.9681 |
| 18 to 21 | 0.9978 | 1.0000 | 0.9967 | 0.9973 | 1.0000 | 0.9434 | 1.0000 | 0.9957 | 1.0000 |
| 20 to 6 | 0.9755 | 0.9911 | 1.0000 | 0.9782 | 0.9016 | 0.9956 | 1.0000 | 0.9578 | 0.9622 |
| 23 to 13 | 0.7642 | 0.7928 | 0.8283 | 0.7698 | 0.8945 | 0.8534 | 0.8491 | 0.8867 | 0.8531 |
| 24 to 12 | 0.9396 | 0.9941 | 0.9706 | 0.9967 | 0.9810 | 0.9943 | 0.9785 | 0.9645 | 0.9844 |

# E  Datasets

**HHAR:** Stisen et al. (2015)
License: CC BY 4.0
https://archive.ics.uci.edu/dataset/344/heterogeneity+activity+recognition

**WISDM:** Kwapisz et al. (2011)
License: CC BY 4.0
https://archive.ics.uci.edu/dataset/507/wisdm+smartphone+and+smartwatch+activity+and+biometrics+dataset

**UCIHAR:** Anguita et al. (2013)
License: CC BY 4.0
https://archive.ics.uci.edu/dataset/240/human+activity+recognition+using+smartphones

**PXECG:** Wagner et al. (2020)
License: CC BY 4.0
https://physionet.org/content/ptb-xl/1.0.3/

# F  Statistical Significance of results

We repeated our experiment over 5 random seeds and reported the mean number in the main paper. The standard deviations for our method can be found below:

Table A15: **Mean accuracy and macro F1 scores on timeseries domain adaptation benchmarks over 5 runs .**

| Method | Mean Shift | | UCIHAR | | HHAR | | PXECG | | WISDM | | WISDM-Bal | |
|---|---|---|---|---|---|---|---|---|---|---|---|---|
| | ACC | F1 | ACC | F1 | ACC | F1 | ACC | F1 | ACC | F1 | ACC | F1 |
| Supervised | 43.12 ±2.03 | 0.423 ±0.019 | 77.04 ±4.09 | 0.750±0.061 | 59.40±4.14 | 0.543±0.045 | 63.51±5.29 | 0.366 ±0.141 | 64.90±4.51 | 0.504±0.066 | 65.84 ± 4.75 | 0.521± 0.073 |
| DANN | 71.32 ±4.55 | 0.701 ±0.054 | 82.91 ±5.98 | 0.857±0.078 | 71.27±6.37 | 0.678±0.061 | 62.87±5.08 | 0.347 ±0.112 | 67.94±5.60 | 0.567±0.081 | 73.86 ± 6.81 | 0.683 ± 0.105 |
| AdvSKM | 74.31 ±4.71 | 0.712 ±0.061 | 85.12 ±6.10 | 0.813±0.079 | 63.25±4.91 | 0.616±0.040 | 62.98±5.11 | 0.372 ±0.127 | 69.92±5.94 | 0.581±0.083 | 71.19 ± 6.24 | 0.611 ± 0.086 |
| CoDATS | 54.31 ±4.18 | 0.531 ±0.0404 | 86.34 ±6.39 | 0.856±0.084 | 68.79±5.71 | 0.686±0.048 | 66.30±5.82 | 0.366 ±0.104 | 68.35±6.11 | 0.548±0.062 | 75.15 ± 8.91 | 0.665 ± 0.107 |
| CDAN | 79.54 ±7.61 | 0.813 ±0.081 | 84.59 ±6.01 | 0.836±0.079 | 70.06±4.19 | 0.704±0.051 | 64.29±5.13 | 0.375 ±0.117 | 70.12±6.11 | 0.517±0.054 | 70.29 ± 8.13 | 0.661 ± 0.092 |
| SASA | 63.72 ±6.03 | 0.587 ±0.071 | 80.75 ±5.93 | 0.791±0.071 | 65.85±3.67 | 0.641±0.039 | 66.47±6.22 | 0.401 ±0.123 | 67.60±5.94 | 0.564±0.057 | 82.81 ± 7.23 | 0.781 ± 0.103 |
| DeepCoral | 82.34 ±5.92 | 0.841 ±0.147 | 86.53 ±7.11 | 0.851±0.083 | 66.16±3.91 | 0.690±0.047 | 62.60±5.15 | 0.346 ±0.143 | 72.72±6.11 | 0.605±0.063 | 74.31 ± 6.74 | 0.649 ± 0.063 |
| CLUDA | 78.21 ±7.39 | 0.802 ±0.132 | 82.45 ±6.25 | 0.854±0.086 | 67.03±3.78 | 0.641±0.032 | 64.92±6.11 | 0.324 ±0.165 | 65.57±7.17 | 0.504±0.068 | 73.77 ± 6.29 | 0.699 ± 0.067 |
| SinkDiv | 73.11 ±7.16 | 0.713 ±0.631 | 85.13 ±4.46 | 0.876±0.048 | 69.64±4.12 | 0.720±0.083 | 64.97±5.62 | 0.376 ±0.074 | 67.16±8.37 | 0.578±0.096 | 70.98 ± 6.83 | 0.648 ± 0.112 |
| RAINCOAT | 73.11 ±7.16 | 0.713 ±0.631 | 89.13 ±6.23 | 0.873±0.081 | 62.11±6.27 | 0.603±0.071 | 66.22±5.52 | 0.357 ±0.032 | 62.11±8.12 | 0.523±0.112 | 69.09 ± 9.29 | 0.727 ± 0.102 |
| SSSS-TSA | 99.01 ±5.04 | 0.985 ±0.041 | 90.12 ±4.01 | 0.901±0.051 | 72.19±4.12 | 0.737±0.047 | 66.38±4.37 | 0.419 ±0.031 | 75.19±8.54 | 0.635±0.091 | 83.57 ± 7.14 | 0.816 ± 0.091 |

