# OpenReview forum: "Time Series Domain Adaptation via Channel-Selective Representation Alignment"
_TMLR — Accepted by TMLR_

### Review · Reviewer_cSVh · 2024-11-07

**Summary Of Contributions:**

This paper proposes a method to address the problem of time series domain adaptation. Specifically, it aligns individual channels and weighted channels through optimal transport. The proposed method is meaningful.

**Audience:**

Yes

**Claims And Evidence:**

No

**Requested Changes:**

See Weaknesses.

**Strengths And Weaknesses:**

**Strengths**:

1. The method achieves good performance across multiple datasets.
2. The paper is well-organized, and the content is relatively complete.

**Weaknesses**:

1. As far as I know, the Sinkhorn algorithm is highly sensitive to hyperparameters. Please provide relevant analysis and strategies for parameter selection.
2. Please provide guidance on how to choose C and the basis for channel division, as well as the impact of these choices.
3. Typically, the Sinkhorn algorithm performs iterations, which can increase computational cost. Please provide time complexity analysis and experimental evidence. Additionally, it would be helpful to include a convergence analysis.
4. The motivation section mentions that the design primarily considers channel information and stability; please add relevant experimental analysis to support this.
5. Please provide an analysis of the network structure's impact on the results.
6. Add more implementation details, such as dataset splitting.
7. Please carefully review the paper as there are some editorial mistakes, for example, in the reference ‘Contrastive Learning for Unsupervised Domain Adaptation of Time Series.’

---

> ### Author Response · Authors · 2024-12-06
> **Response to Reviewer cSVh**
>
> We would like to thank Reviewer csVh for acknowledging the strengths of our paper. We are also very thankful for your constructive comments and questions that have helped us improve our manuscript.
>
> We address these comments in detail below:
>
> **1. As far as I know, the Sinkhorn algorithm is highly sensitive to hyperparameters. Please provide relevant analysis and strategies for parameter selection**
>
> **Response**:  Thank you for raising this important point. We have now added discussion and results that analyze sensitivity to Sinkhorn divergence hyperparameters in Appendix C.6.1.
> Table 1R below  provides these results for how changing the entropic regularization parameter $\gamma$ changes results on 3 channel WISDM-Bal and 9 channel UCIHAR  datasets.
> | $\gamma$ | WISDM-Bal | UCIHAR |
> | ----------- | --------- | ------ |
> | 1e-4      | 0.8807    | 0.914  |
> | 1e-3       | 0.8571    | 0.915  |
> | 1e-2        | 0.8641    | 0.889  |
> | 1e-1         | 0.915     | 0.883  |
> | 1           | 0.708     | 0.781  |
> | 10          | 0.691     | 0.701  |
>
> **Table 1R: . Macro F1 scores on WISDM-Bal and UCIHAR datasets as $\gamma$ is varied**
>
>  For a $\gamma$ value greater than and equal to 1, there is a sharp drop off in results across all datasets. These are expected results as for higher values of $\gamma$, the  transport plan smooths out. This causes  Sinkhorn divergence to lose its geometric discerning properties which can affect performance.
> As target labels are not available for domain adaptation, we do not tune the gamma parameter. We set it to 0.001 for all experiments. We do this similar to [2], in which $\gamma$ is set to 0.001.
>
> **2. Please provide guidance on how to choose $C$ and the basis for channel division, as well as the impact of these choices.**
>
> **Response**: $C$ is the number of input channels present in input data. Thus if the the input data has $N$ channels/dimensions, $C$ would be set to $N$. The example provided in figure 2 shows input data with 3 channels/dimensions (Blue, Magenta, and Green). Thus $C$ is set to 3 here.
>
> **3. Typically, the Sinkhorn algorithm performs iterations, which can increase computational cost. Please provide time complexity analysis and experimental evidence. Additionally, it would be helpful to include a convergence analysis.**
>
> **Response**: To better analyze the effect of Sinkhorn iterations and computation cost, we have added discussion and results in subsection Appendix C.6.2. As you mention, a large number of iterations can increase computational cost.  Table 2R below shows the number of iterations required for different values of $\gamma$ at different  $\epsilon$ (tolerance) values for the Sinkhorn potentials.
>
> | $\gamma$ | $\epsilon =0.001$ | $\epsilon =0.1$ |
> | ------| --------------- | ---------------- |
> | 1e-4  | 734             | 5                |
> | 1e-3  | 494             | 5                |
> | 1e-2  | 274             | 5                |
> | 1e-1  | 59              | 4                |
> | 1       | 8               | 3                |
>
> **Table 2R: Macro F1 scores for different different   $\epsilon$  iteration tolerances at different values of $\gamma$**
>
>  For epsilon value of 1e-3, we can see that the number of iterations is very large when $\gamma$ values are small.
> One commonly used approach to speed up Sinkhorn divergence with minimal performance degradation, is to relax  the $\epsilon$ tolerance. This has been explored in  [1] where the authors show on simulated data  how only 5 iterations at a smaller values of $\gamma$ can provide a good balance between performance and computation speed.  An epsilon value of 1e-1 was also used by other domain adaptation papers [2] .
> For these reasons, we used an epsilon value of 1e-1 for all our experiments in our paper (including Table 1R results above). An $\epsilon$ value of 1e-1 results in a lower number of required iterations. This provided us the right balance between computation efficiency and performance.
>
> On the  WISDM dataset with 3 input channels, the time   needed to compute sinkhorn divergence between each source and target channel  representations  was about 0.00135 seconds at this epsilon value of 1e-1 and $\gamma$ value of 1e-3.
>
> A more  theoretical convergence analysis is perhaps beyond the scope of this paper, but there are many useful works in the literature that analyze this convergence [3]. We have cited these in our paper now.
>
> [1] Bigot, Jérémie, et al. "On the potential benefits of entropic regularization for smoothing Wasserstein estimators." arXiv preprint arXiv:2210.06934 (2022).
>
> [2] .He, Huan, et al. "RAINCOAT: Domain adaptation for time series under feature and label shifts." International Conference on Machine Learning. PMLR, 2023.
>
> [3] Ghosal, Promit, and Marcel Nutz. "On the convergence rate of sinkhorn's algorithm." arXiv preprint arXiv:2212.06000 (2022).

---

> > ### Author Response · Authors · 2024-12-06
> > **Continued: Response to reviewer comments cSVh**
> >
> > **4. The motivation section mentions that the design primarily considers channel information and stability; please add relevant experimental analysis to support this.**
> >
> > **Response**: By channel information and stability, we mean the information contained within each channel could be very important for classification tasks. Corruption within some channels could corrupt class  information contained in other channels.  Channel specific classifiers help preserve channel level class information.
> >
> > To better analyze this, we have  ablations that remove channel level classifiers in Table 2 (on page 9) . This ablation ultimately leads to a model that doesn’t consider channel level class information. Table 2, (W/O Ind Classifier) , shows results for this setting where channel level class information is discarded. This leads to a significant drop in performance (performance drops by 37% ) on the UCIHAR dataset. We also make the connection between these results and channel information in the third paragraph of ablation section 4.5 (Last paragraph on Page8: How important is the classifier for each channel).
> >
> > **5. Please provide an analysis of the network structure's impact on the results.**
> >
> > **Response**: We have included ablations in Table  2 (Page 9) that  analyze how different components of a network affect results. We study how performance drops by 15 % when our proposed attention layer is removed. This is further analyzed in channel level corruption results in Figure 4. Our ablation results which remove channel specific classifiers also analyze the network structure’s impact on results.  We also analyze in Appendix Table A1 how the performance is affected when the same encoder is used for aligning all channels.
> >
> > **6. Add more implementation details, such as dataset splitting.**
> >
> > **Response**: As we use existing standardized datasets for domain adaptations. these datasets have designated training and testing splits. We mention this in Appendix B, but have now made it more explicit for improved clarity. We have also included details for validation splitting for results that require validation held out sets.
> >
> >
> > **7. Please carefully review the paper as there are some editorial mistakes, for example, in the reference ‘Contrastive Learning for Unsupervised Domain Adaptation of Time Series.**
> >
> > **Response**:   Thank you for pointing this out. We have corrected this. We  also plan to conduct a thorough careful review of all editorial details after additional  changes/edits are made at the end of  this discussion period.

---

> > > ### Author Response · Authors · 2025-01-02
> > > **Your feedback on our replies above**
> > >
> > > Dear reviewer cSVh,
> > >
> > > Thank you again for your constructive questions and comments.
> > >
> > > We have tried addressing your comments and questions in our replies above.
> > > We are eager to hear your feedback and discuss any remaining concerns.
> > >
> > > Thank you

---

### Review · Reviewer_2MS2 · 2024-11-11

**Summary Of Contributions:**

The submission introduces a novel approach to **domain adaptation for multivariate time series** called **Signal Selection and Screening via Sinkhorn alignment for Time Series domain Adaptation (SSSS-TSA)**. This method addresses the limitations of existing approaches by adapting to channel-specific shifts and variances in multivariate time series data, which are common in real-world scenarios but often overlooked in current domain adaptation techniques.

**Audience:**

Yes

**Claims And Evidence:**

Yes

**Requested Changes:**

see Weaknesses

**Strengths And Weaknesses:**

### Strong Aspects of the Submission

1. **Novel Approach to Channel-Specific Domain Adaptation**:
   - The **SSSS-TSA method** introduces a highly innovative strategy for selectively aligning and weighting channels in multivariate time series data, addressing limitations in existing domain adaptation approaches that overlook channel-specific shifts.

2. **Comprehensive Evaluation**:
   - Extensive experiments across a variety of real-world datasets, including synthetic and corrupted data, demonstrate **robustness and adaptability** in challenging scenarios with different types of domain shifts (e.g., noise, saturation, and dropped channels).
   - The method consistently outperforms competitive baselines in terms of accuracy and macro F1 scores, showcasing **state-of-the-art performance**.

3. **Interpretability and Practical Applicability**:
   - The **channel selection and screening mechanism** enhances interpretability, providing insights into the channels that contribute most to domain adaptation performance, which is particularly valuable for applications needing diagnostic insight (e.g., healthcare or sensor-based monitoring).

4. **Mathematical Rigor with Efficient Implementation**:
   - The use of **Sinkhorn divergence** for alignment provides a theoretically grounded and computationally efficient way to handle the distribution shifts, with adaptive scaling for better convergence in the alignment process.

### Weaker Aspects Requiring Attention

1. **Complexity of Implementation and Scalability**:
   - The method’s architecture, involving separate encoders and classifiers for each channel, may lead to **increased computational complexity** and memory usage, especially for high-dimensional time series data with many channels. A detailed discussion on the scalability of the approach for large datasets or high-dimensional inputs could improve understanding and practicality for deployment.

2. **Ablation Studies on Attention Mechanism**:
   - While the **channel selection and screening** mechanism is shown to be effective, a more detailed exploration of different **attention mechanisms or alternative weighting schemes** would strengthen the empirical evidence and offer insights into the method’s sensitivity to variations in the attention layer.

3. **Sensitivity Analysis of Hyperparameters**:
   - The model’s effectiveness depends on certain hyperparameters, such as the **Sinkhorn divergence entropy parameter \( \gamma \)** and **softmax scaling factor \( \tau \)**. Additional results on the **sensitivity of the method to these hyperparameters** would aid practitioners in tuning the model for optimal performance across different datasets.

4. **Potential Overfitting in Highly Noisy Settings**:
   - Given the focus on channel-level alignment, there may be a risk of **overfitting in scenarios with extreme noise or highly sparse data**. Further insights or regularization techniques to mitigate potential overfitting in such scenarios would be beneficial.

5. **Generalizability Beyond Time Series**:
   - While the approach is specifically designed for time series, there may be potential to apply channel-specific domain adaptation principles to other **multimodal or spatial-temporal data** types. A brief discussion on extending the method to broader settings would enhance the contribution's appeal and relevance.

### Suggestions for Improvement

- Adding a **scalability analysis** section or extending the discussion on computational efficiency could provide clarity on the method’s usability in larger or more complex datasets.
- **Exploring alternative channel weighting mechanisms** or conducting a more granular comparison across different attention mechanisms would reinforce the robustness of the channel screening module.
- Providing further **guidelines for hyperparameter tuning**, particularly regarding \( \gamma \) and \( \tau \), would make the method more accessible and reproducible for other researchers and practitioners.

Overall, the submission offers a highly valuable contribution to domain adaptation in time series, with potential for future expansion and refinement to enhance scalability, robustness, and adaptability across even more varied data contexts.

---

> ### Author Response · Authors · 2024-12-06
> **Response to Reviewer 2MS2**
>
> Reviewer 2MS2, thank you so much for recognizing the strengths of our proposed work. We are also very thankful to your detailed constructive comments that have helped us improve our manuscript.
>
> We address these comments in detail below:
>
> **1. The method’s architecture, involving separate encoders and classifiers for each channel, may lead to increased computational complexity and memory usage, especially for high-dimensional time series data with many channels. A detailed discussion on the scalability of the approach for large datasets or high-dimensional inputs could improve understanding and practicality for deployment.**
>
> **Response**: Thank you for raising this important point. We have now included section C.5. in the Appendix which addresses computation complexity and memory usage. Yes, having a separate encoder and classifier for each channel does lead to increased computational and memory usage. Though as  time series encoders such as 1DCNN are much smaller in size  than vision or language encoders, the increased memory demands are manageable. For all our experiments in our results, we used an encoder that consisted of 3 layers of 1DCNN. The size of each of these channel specific encoders was  0.38014 MB.  Each channel specific classifier was of size 0.065MB. Such a small memory footprint would allow our model  to adequately handle memory usage for high dimensional input time series datasets.
> These channel specific encoders and classifiers also have manageable computational complexity. For the 9 channel UCI HAR dataset,   time taken for a batch of 64 samples to forward pass and backpropagate in total is  0.109 seconds.
>
>  For higher dimensional time series, multiple GPUs and multithreading for different encoders can be used to further speed up computations and improve time complexity.
>
> **2. While the channel selection and screening mechanism is shown to be effective, a more detailed exploration of different attention mechanisms or alternative weighting schemes would strengthen the empirical evidence and offer insights into the method’s sensitivity to variations in the attention layer.**
>
> **Response**: Obtaining an effective attention scheme has been a challenging part of our project. We discuss alternate attention mechanisms in Appendix C.2.  We initially considered self-attention (and related multiheaded attention ), but the results for this mechanism were subpar as shown in Table A2. The self-attention mechanism computes attention through interactions between all possible pairs of inputs. In our case, these inputs would be different channels. If some of these channels are corrupted, then these interactions would be corrupted too, leading to subpar performance. For this reason we introduce a variant of attention which constraints channels weightings to not involve interactions between channels. These are discussed in more detailed in section Appendix C.2.
>
> **3. The model’s effectiveness depends on certain hyperparameters, such as the Sinkhorn divergence entropy parameter ($ \gamma$ ) and softmax scaling factor ( $\tau$ ). Additional results on the sensitivity of the method to these hyperparameters would aid practitioners in tuning the model for optimal performance across different datasets.**
>
> **Response**: We have added sensitivity results and discussion for different hyperparameters in a new Appendix section C.6.  The discussion in the sections can guide practitioners on how to tune parameters. Table A4  in the manuscript  (Table 1R in response to reviewer  cSVh)  shows how performance is affected when $\gamma$ parameter is changed. We also study the relation between $\gamma$ parameter and the number of iterations needed for different tolerance values in Table A5.
>
> We also provide detailed discussion results for sensitivity to $\tau$ in Appendix C.6.3 and Table A6 in the manuscript (which are also provided in Table 3R below).
> | $\tau$ | WISDM (20 to 30) | UCI_HAR (12 to 16) |
> | ----- | ---------------- | ------------------ |
> | 0.1   | 0.884            | 0.807              |
> | 1     | 0.896            | 0.858              |
> | 3     | 0.891            | 0.851              |
> | 5     | 0.898            | 0.855              |
> | 10    | 0.898            | 0.862              |
> | 20    | 0.897            | 0.87               |
> | 50    | 0.878            | 0.852              |
>
> **Table 3R: Sensitivity to $\tau$. F1 scores on WISDM-Bal and UCIHAR**

---

> ### Author Response · Authors · 2024-12-06
> **Response to Reviewer 2MS2 (continued)**
>
> Table 3R above shows the  the effect of varying $\tau$  on the 3 channel WISDM-Bal dataset and the 9 channel UCIHAR dataset.
> We can see that the results are mostly similar across different values of $\tau$, except for $\tau$ = 0.1 on the 9 channel UCIHAR dataset.   For smaller values of $\tau$ , the channel selection weights provided by the softmax distribution are encouraged to have a low entropy. This results in much larger weights to be assigned to a few channels, while most other channels are ignored. As with other hyperparameters, target labels are often not available in real-world domain adaptation settings. For this reason we set a rule  for selecting the $\tau$ parameter. equal to the number of channels, so that entropy of the channel selection scheme scales with the number of input channels. Though the correct temperature value would depend on the channel informativeness/ and use case practitioners are interested. If practitioners think that almost all channels  should be ignored and very few channels should be selected (and many channels are noisy), then a small value of $\tau$ would be suitable. If practitioners think only few channels should be dropped then a larger value of $\tau$ would be preferable. Classification performance on source domain validation held-out labels could be a good indicator for determining which of this is more desirable.
>
> **4. “Given the focus on channel-level alignment, there may be a risk of overfitting in scenarios with extreme noise or highly sparse data. Further insights or regularization techniques to mitigate potential overfitting in such scenarios would be beneficial.**
>
> This is quite possible. We think that the channel specific classifiers  can  help identify and focus on non-noisy/more informative channels in the source domain channels for which labels are available through held out validation set. Such validation set on source labels can  help practitioners design regularization schemes for channel level classifiers to avoid overfitting. We can't say much about the target domain where target labels are not available. Though the hope is that the regularized/informative channel level alignment classifiers that are obtained through source labels could perhaps also improve performance on target domain channels through the alignment loss.
>
>
> **5.While the approach is specifically designed for time series, there may be potential to apply channel-specific domain adaptation principles to other multimodal or spatial-temporal datatypes. A brief discussion on extending the method to broader settings would enhance the contribution's appeal and relevance.**
>
> **Response**: Thank you for this very constructive suggestion. We have added text in the discussion section (in blue) on how there is room to extend our work to multi-modal domain adaptation settings in last paragraph of section  5 (Conclusion and Future Directions). Thank you again for this.

---

> > ### Author Response · Authors · 2025-01-02
> > **Your feedback on our replied above**
> >
> > Dear reviewer 2MS2,
> >
> > Thank you again for your constructive questions and comments.
> >
> > We have tried addressing your comments and questions in our replies above. We are eager to hear your feedback and discuss any remaining concerns.
> >
> > Thank you

---

### Review · Reviewer_q8uh · 2024-11-20

**Summary Of Contributions:**

*Idea*: The paper presents *SSSS-TSA* (Signal Selection and Screening via Sinkhorn alignment for Time Series domain Adaptation), a novel approach for time series domain adaptation. The method addresses channel-wise shifts by aligning both individual channel representations and global representation while reweighting the channels to take into account important ones and potentially discarding channels that present large shifts.

*Motivation*: The main motivation behind the approach is to make use of the channel-wise information and notably to discard channels where the shift is too large, which may hinder the global representation and alignment.

*Method*: The authors rely on an optimal transport loss, namely the Sinkhorn divergence, to align both individual and global representation and on a usual classification loss for the channel-level and global-level classification (e.g., cross-entropy loss). The channel screening and selection process (via reweighting) is performed on the individual encoder representation of the channels and is akin to the attention mechanism (with query and key matrices). The global representation consists of the vectorized reweighted individual representations.

*Experiments*: The authors conduct extensive experiments on usual time series domain adaptation benchmarks as well as the WISDM human activity dataset. Ablation studies on the method and the shift present in the data are also provided. The authors demonstrate superior performance (often significantly) over their competitors. The proposed approach also enables some interpretablity regarding the individual channel shifts.

**Audience:**

Yes

**Broader Impact Concerns:**

No ethical concerns.

**Claims And Evidence:**

Yes

**Requested Changes:**

- Could the authors add in bold the best results in Table 2 for the sake of clarity?
- It seems that there is a missing ref on page 7 in the "baselines" paragraph for DeepCoral.
- Coudl the author add a discussion/comparison on the computational efficiency of SSSS-TSA?

**Strengths And Weaknesses:**

**Strengths**
- The paper is well written, with clear notations, and motivations;
- The proposed approach is smart and very efficient;
- The choice of baselines is comprehensive and covers well the literature;
- The experiments are large-scale and the ablation studies are very complete;
- The results look impressive where SSSS-TSA is superior to the state-of-the-art competitors and by far;
- The approach seems very robust to channel corruption;
- I particularly appreciate the honesty and scientific integrity of the authors (see experiments and evaluation details paragraph) that evaluate their approach **after** all the epochs instead of using a held-out validation set that **does not exist in real applications**. I believe this is the good approach to domain adaptation and to semi-supervised learning in general that deserves to be more used in the community.

**Weaknesses**
- A computational efficiency comparison is missing or at least a discussion on that. Could the authors provide some discussion/comparison in terms of training time and memory efficiency? I wonder if and how much having the individual classification and the optical transport loss hinder the computational efficiency.

**Questions**
- I am a little bit confused by the examples of Figure 1 where there seems to be a channel permutation between the source and the target. Could the authors give real applications where this happens? If it does, then how do we distinguish between channel 1, chennl 2 and channel 3 to perform encoding and alignment at the channel-level explained in Figure 2?

**A Kind Opinion**
- I wonder whether the acronym "SSSS-TSA" is the best choice. Usually, people tend to use a power to avoid repetition like *$S^4$-TSA* that also has the benefit of being pronounced like "[s] [for] [t][s][a]", which is simpler to pronounce that "[s][s][s][s] - [t][s][a]". This is simply an opinion and I am not judgmental on your acronym choice.

Overall, I find the paper particularly interesting and solid and would strongly recommend an acceptance.

---

> ### Author Response · Authors · 2024-12-06
> **Response to Reviewer q8uh**
>
> Reviewer q8uh, we are very grateful for your positive comments which highlight the strengths of our work. We are glad you see value in reporting results after all epochs have elapsed (as compared to using a held-out validation set)!
>
> We are also very thankful for your constructive comments and suggestions that have helped us improve our manuscript.
>
> We address these comments  in detail below.
>
> **1. A computational efficiency comparison is missing or at least a discussion on that. Could the authors provide some discussion/comparison in terms of training time and memory efficiency? I wonder if and how much having the individual classification and the optical transport loss hinder the computational efficiency.**
>
> **Response**: Thank you for this constructive suggestion which has also been echoed by other reviewers. We have also now added a section in Appendix C.5. that discusses the increase in training time and memory utilization when separate encoders, classifiers, and transport plans are used for each channel. We also analyze how computational and memory demands increase when using a channel specific encoder/classifier network, as compared to a single encoder, network architecture. The computational and memory needs certainly are larger (scale slightly lesser than the number of channels) than  when a single encoder/classifier is used. However, each of these channel specific encoders is very small (0.36MBs).The total time taken for a batch of size 64 on the 9 channel UCIHAR dataset to forward pass and backpropagate to update all parameters for our method is 0.109 seconds. This make the computation and memory demands of our model manageable. There is certainly room to improve this further by utilize parallel computation through multiple GPUs/multithreading  for channel specific encoders.
>
> **2.  I am a little bit confused by the examples of Figure 1 where there seems to be a channel permutation between the source and the target. Could the authors give real applications where this happens? If it does, then how do we distinguish between channel 1, chennl 2 and channel 3 to perform encoding and alignment at the channel-level explained in Figure 2?**
>
> **Response**:  We apologize for the confusion related to this. The offset in the overall positioning  between the source and the target figures mistakenly gave the impression that there are channel permutations.The blue channel between source and target channels has a large shift, whereas the orange and green channels are relatively the same across source and target.  We have revised these figures to eliminate these offsets so that these figures do not give the impression of channel permutation. Thank you for pointing this out. These figures are from the HHAR_SA dataset - other examples can be seen in Figure 5.
>
> **3. Could the authors add in bold the best results in Table 2 for the sake of clarity?**
>
> **Response**: Thank you for this suggestion. We have added the best results in bold now.
>
> **4. It seems that there is a missing ref on page 7 in the "baselines" paragraph for DeepCoral.**
>
> **Response**: Thank you for bringing this to our notice. We have added this missing reference now.
>
>
> **5.  wonder whether the acronym "SSSS-TSA" is the best choice. Usually, people tend to use a power to avoid repetition like -TSA that also has the benefit of being pronounced like "[s] [for] [t][s][a]", which is simpler to pronounce that "[s][s][s][s] - [t][s][a]". This is simply an opinion and I am not judgmental on your acronym choice.**
>
> **Response**: Thank you for this suggestion and your kind thoughts which are very constructive.
> We chose the title "SSSS-TSA" for our channel selection and screening domain adaptation method, as "[SSSS](https://en.wikipedia.org/wiki/Secondary_Security_Screening_Selection)" is also the acronym for Secondary Security Screening Selection which is used by the Transport Security Administration (TSA) to further screen international air travellers.  We thought the channel selection and screening layer for our method can be a subtle nod to  this other security screening selection (which authors of this paper, and many other international graduate students, are accustomed to). Though we are certainly open to suggestions and are again thankful to you for your constructive comments.

---

> > ### Comment · Reviewer_q8uh · 2024-12-06
> > **Thank You!**
> >
> > I thank the authors for their detailed answers and the additional experiments and discussion in the revised manuscript.
> >
> > My concerns on the channel permutation and computational cost have been addressed and I maintain my acceptance recommendation as I think the paper is relevant for the time series/domain adaptation community. In my opinion, the increase in computational cost is balanced by the significant performance improvement of the method, although further work should/could be done to alleviate this bottleneck.
> >
> > I was unaware of this 'SSSS' acronym and I thank the authors for their explanation.

---

### Author Response · Authors · 2024-12-06
**General response to reviewer comments**

Thank you to all reviewers for their comments that our method “achieves good performance across multiple datasets.” (cSVh) and that the “paper is well-organized, and the content is relatively complete.” (cSVh); “highly innovative strategy for selectively aligning and weighting channels in multivariate time series data” (2MS2); “choice of baselines is comprehensive and covers well the literature.” (q8uh) “The experiments are large-scale and the ablation studies are very complete” (q8uh)

We also are very grateful to the detailed constructive feedback, comments and questions that have helped us improve our manuscript.


In response to these reviewers, we have added additional results and discussion  in the updated manuscript. These additions can be seen in blue.

We have conducted additionally parameter sensitivity analysis of the Sinkhorn regularization parameter $\gamma$ (cSVh, 2MS2) and the attention temperature parameter $\tau$ in the new appendix section C6.. We also provide an analysis of the time complexity of the Sinkhorn iterations  (cSVh, and 2MS2, q8uh) in section C6.2.

To better  address the computational  complexity, we have added  a new section (C.5) in the Appendix which discusses time and memory complexity. The main summary of these results is that each channel encoder is very lightweight (about 0.36MBs). This makes memory and time complexity relatively manageable as the number of channels increase.

We provide detailed responses to individual questions and comments below:

---

### Decision · Action_Editor_z3Ar · 2025-01-14

**Recommendation:** Accept as is

**Comment:**

This manuscript presents a reasonable approach to time series domain adaptation, which exploits a basic idea of representation alignment on selective channels. This design will benefit practical scenarios where OOD happens differently on different channels. The adoption of the Sinkhorn algorithm is relatively new in domain adaptation. The experimental results are strong compared to state-of-the-art methods, and the evaluation methodology without tuning on a validation set is highly appreciated by some reviewers.

**Audience:**

Readership in domain adaptation (machine learning) and time series analysis (data science) would be interested in this paper when they are applying existing methods in OOD scenarios.

**Claims And Evidence:**

Most of the claims made in the submission are supported by clear evidence. The technical innovations claimed in the paper are not fully received by the reviewers, but it appears acceptable because the merits presented in the paper outweigh the weaknesses.